# Spectral Analysis of Representational Similarity with Limited Neurons

**Hyunmo Kang**[*]
Department of Physics
Johns Hopkins University
Baltimore, MD 21218
hkang56@jh.edu

**Abdulkadir Canatar**[*]
Center for Computational Neuroscience
Flatiron Institute
New York, NY 10010
acanatar@flatironinstitute.org

**SueYeon Chung**
Department of Physics
Harvard University
Cambridge, MA 02138
sueyeonchung@g.harvard.edu

## Abstract

Understanding representational similarity between neural recordings and computational models is essential for neuroscience, yet remains challenging to measure reliably due to the constraints on the number of neurons that can be recorded simultaneously. In this work, we apply tools from Random Matrix Theory to investigate how such limitations affect similarity measures, focusing on Centered Kernel Alignment (CKA) and Canonical Correlation Analysis (CCA). We propose an analytical framework for representational similarity analysis that relates measured similarities to the spectral properties of the underlying representations. We demonstrate that neural similarities are systematically underestimated under finite neuron sampling, mainly due to eigenvector delocalization. Moreover, for power-law population spectra, we show that the number of localized eigenvectors scales as the square root of the number of recorded neurons, providing a simple rule of thumb for practitioners. To overcome sampling bias, we introduce a denoising method to infer population-level similarity, enabling accurate analysis even with small neuron samples. Theoretical predictions are validated on synthetic and real datasets, offering practical strategies for interpreting neural data under finite sampling constraints.

## 1 Introduction

Understanding how artificial neural networks relate to biological neural activity remains one of the central challenges in computational neuroscience [13, 50, 40]. As deep learning models become increasingly sophisticated at matching human-level performance on complex tasks, there is growing interest in whether these models actually learn representations that mirror those found in the brain [53, 26, 25, 45, 35]. However, a fundamental obstacle stands in the way of making this comparison: while artificial networks can be analyzed in their entirety, neuroscientists can only record from a small subset of neurons in any given brain region [10, 51, 47]. This sampling limitation poses a critical challenge for the field. When we measure the similarity between model and neural representations using standard techniques like Canonical Correlation Analysis (CCA) or Centered Kernel Alignment (CKA), how much does our limited neural sample size distort the true relationship? Addressing this

---

[*]Equal contribution

39th Conference on Neural Information Processing Systems (NeurIPS 2025).

issue is critical, given that these metrics increasingly inform model selection and neuroscientific interpretation [42, 39].

Our work provides the first rigorous theoretical framework for understanding how neuron sampling affects representational similarity measures. Our analysis reveals that measuring CCA and CKA with a limited number of recorded neurons systematically underestimates the true population-level similarity. This underestimation stems primarily from eigenvector delocalization [1, 15, 4]—a phenomenon where sample eigenvectors become increasingly misaligned with their population counterparts as the number of recorded neurons decreases.

Our analysis proceeds in two parts. First, in the forward problem, we investigate how neuron sub-sampling from the full underlying population distorts the population eigencomponents and how this distortion affects the computed similarity measures. Second, in the backward problem, we ask whether observations from a finite number of neurons can be used to reliably infer the population representational similarity.

## 1.1  Our Contributions

- **Eigencomponent-wise Analysis of Representation Similarity:** We show how neuron sub-sampling alters the eigenvalues and eigenvectors of the Gram matrix, leading to a systematic underestimation of CCA/CKA due to eigenvector delocalization.

- **Backward Inference via Denoising Eigenvectors:** We introduce a denoising method that leverages population eigenvalue priors (e.g., power-law) to infer the true population similarity from limited data, substantially correcting the sampling bias.

- **Validation on Real Neural Data:** Applying our framework to primate visual cortex recordings confirms that even modest neuron counts can lead to severe underestimation of model–brain similarity and that our method effectively recovers the missing signal.

## 1.2  Related Works

Representation similarity measures expressed in terms of eigencomponents were presented in detail by Kornblith et al. [29], who showed that CCA, CKA, and linear regression scores can all be written in terms of the eigenvalues and eigenvectors of the Gram matrices.

A key question is how these similarity measures behave under different kinds of noise. Broadly, there are two primary noise sources:

1. *Additive noise*, which arises from trial-to-trial variability and measurement error. In many studies, repeated trials and averaging can substantially mitigate this type of noise.

2. *Sampling noise*, which occurs because we can only record from a limited subset of neurons rather than the entire population. Consequently, the sample eigenvectors and eigenvalues differ from their population counterparts.

In this work, we focus on the latter issue—sampling noise—since we assume trial averaging already reduces the additive noise to a manageable level.

One approach to address sampling noise is by studying the *moments* of the Gram matrix [28, 14]. While these methods provide a way to approximate the effect of sampling on the scalar values of certain similarity measures, they do not directly offer an interpretable description of what happens to the underlying eigencomponents. Recent work by [41] provides bounds on representation similarity measures when the number of sampled neurons is limited. However, these bounds are tight only under the assumption of a white Wishart model (i.e., all population eigenvalues are 1). For more realistic data, where eigenvalues often decay according to a power-law(primary visual cortex for mice in [49] and human visual cortex fMRI in [20]), these bounds can become too loose to be practically informative.

Instead, we directly investigate how sampling noise affects both the eigenvalues and eigenvectors of the sample Gram matrix using random matrix theory [43, 7, 8]. Extensive results exist for white Wishart matrices and low-rank "spiked" models, including the Baik–Ben Arous–Péché (BBP) phase transition [4], which reveals that sample eigenvectors often serve as poor estimators of their population counterparts. These ideas have been extended to canonical correlation analysis [36, 9]. However, the

power-law-like spectra observed in neural data have not yet received comparable attention. Our work attempts to bridge this gap by studying sampling noise in representations with strongly decaying eigenvalues, which are ubiquitous in neural datasets.

## 2  Notation & Problem Setup

We use bold fonts for matrices and bracket notation for vectors[2]. We use a tilde to denote quantities related to their population values.

We consider two centered population activations $\tilde{\mathbf{X}} \in \mathbb{R}^{P \times \tilde{N}_x}$ and $\tilde{\mathbf{Y}} \in \mathbb{R}^{P \times \tilde{N}_y}$ with $\tilde{N}_x$ and $\tilde{N}_y$ neurons, recorded in response to a fixed set of stimuli of size $P$. Centered means that we subtract the column-wise mean. Their corresponding population Gram matrices are given by $\tilde{\mathbf{\Sigma}}_x = \tilde{\mathbf{X}}\tilde{\mathbf{X}}^\top$ and $\tilde{\mathbf{\Sigma}}_y = \tilde{\mathbf{Y}}\tilde{\mathbf{Y}}^\top$ with eigendecomposition:

$$\tilde{\mathbf{\Sigma}}_x = \sum_{i=1}^P \tilde{\lambda}_i \, |\tilde{u}_i\rangle\langle\tilde{u}_i| \,, \quad \tilde{\mathbf{\Sigma}}_y = \sum_{a=1}^P \tilde{\mu}_a \, |\tilde{w}_a\rangle\langle\tilde{w}_a| \,, \tag{1}$$

where $\tilde{\lambda}_i, \tilde{\mu}_a$ and $|\tilde{u}_i\rangle, |\tilde{w}_a\rangle$ are respectively their eigenvalues and eigenvectors, and the eigenvectors are mutually orthogonal, i.e. $\langle\tilde{u}_i|\tilde{u}_j\rangle = \delta_{ij}$ and $\langle\tilde{w}_a|\tilde{w}_b\rangle = \delta_{ab}$.

The sample activations $\mathbf{X} \in \mathbb{R}^{P \times N_x}$ and $\mathbf{Y} \in \mathbb{R}^{P \times N_y}$ are assumed to be generated from the population ones by a random projection $\mathbf{X} = \tilde{\mathbf{X}}\mathbf{R}$ where $\mathbf{R} \in \mathbb{R}^{\tilde{N} \times N}$ is a random matrix with Gaussian i.i.d entries. Their Gram matrices are defined as $\mathbf{\Sigma}_x = \mathbf{X}\mathbf{X}^\top$ and $\mathbf{\Sigma}_y = \mathbf{Y}\mathbf{Y}^\top$ with eigendecomposition:

$$\mathbf{\Sigma}_x = \sum_{i=1}^P \lambda_i \, |u_i\rangle\langle u_i| \,, \quad \mathbf{\Sigma}_y = \sum_{a=1}^P \mu_a \, |w_a\rangle\langle w_a| \,. \tag{2}$$

Random projections serve as an effective approach for sampling high-dimensional data due to their geometry-preserving properties [31] and are a popular method in analyzing neural dynamics from limited recordings [19]. This assumption allows us to treat sample Gram matrices as structured random Wishart matrices (see SI.A.1).

Noting that both population and sample eigenvectors reside in $\mathbb{R}^P$, we define *self-overlap matrices* between sample and population eigenvectors for each representation as

$$Q_{ij}^x := \mathbb{E}[\langle u_i|\tilde{u}_j\rangle^2], \quad Q_{ab}^y := \mathbb{E}[\langle w_a|\tilde{w}_b\rangle^2] \tag{3}$$

and *cross-overlap matrices* between the eigenvectors of two representations as

$$M_{ia} := \mathbb{E}[\langle u_i|w_a\rangle^2], \quad \tilde{M}_{ia} := \langle\tilde{u}_i|\tilde{w}_a\rangle^2 \tag{4}$$

Expectations are over different instances of neuron sampling via random projections. The cross-overlap $\tilde{\mathbf{M}}$ between two population eigenvectors is deterministic, hence does not require averaging.

### 2.1  Common Representational Similarity Measures

Here, we review common representational similarity measures and show that these measures can be expressed in terms of the average quantities presented above.

**Canonical Correlation Analysis (CCA)** is an algorithm that sequentially finds a set of orthonormal vectors $\{\mathbf{a}_\alpha\}$ and $\{\mathbf{b}_\alpha\}$ for which the correlation coefficients $\rho_\alpha = \mathrm{corr}(\mathbf{X}\mathbf{a}_\alpha, \mathbf{Y}\mathbf{b}_\alpha)$ for two matrices $\mathbf{X}, \mathbf{Y}$ are maximized [23]. The squared sum of these coefficients gives the CCA similarity (CCA $= \sum_\alpha \rho_\alpha^2$), and can be expressed in terms of the overlap matrix $M_{ia}$ [5, 29]

$$\mathrm{CCA} = \sum_{i=1}^{N_x} \sum_{a=1}^{N_y} \frac{\langle u_i|w_a\rangle^2}{\min(N_x, N_y)} = \sum_{i=1}^{N_x} \sum_{a=1}^{N_y} \frac{M_{ia}}{\min(N_x, N_y)}. \tag{5}$$

---

[2]For two vectors $\mathbf{a}, \mathbf{b} \in \mathbb{R}^N$, $\langle a|b\rangle$ denotes their inner product ($\mathbf{a}^\top\mathbf{b}$) and $|a\rangle\langle b|$ their outer product ($\mathbf{a}\mathbf{b}^\top$).

CCA is sensitive to perturbations when the condition number of $\mathbf{X}$ or $\mathbf{Y}$ is large [21]. To enhance robustness, Singular Value CCA (SVCCA) performs CCA on the truncated singular values of $\mathbf{X}$ and $\mathbf{Y}$ [44]. In this approach, the sum of the overlap matrix $\mathbf{M}$ is truncated to include only the first few components. To avoid confusion, from now on, we will refer to SVCCA truncated to the top ten components[3] for both $\mathbf{X}$ and $\mathbf{Y}$ as CCA, i.e (SV)CCA $= \frac{1}{10} \sum_{i=1}^{10} \sum_{a=1}^{10} M_{ia}$.

**Centered Kernel Alignment (CKA)** is a summary statistic of whether two representations agree on the (dis)similarity between a pair of examples based on their dot products [16]. CKA is defined as $\frac{\text{Tr} \, \mathbf{\Sigma}_x \mathbf{\Sigma}_y}{\sqrt{\text{Tr} \, \mathbf{\Sigma}_x^2 \, \text{Tr} \, \mathbf{\Sigma}_y^2}}$ and essentially measures the angle between two Gram matrices. In terms of spectral components, it can be expressed as:

$$\text{CKA} = \sum_{i=1}^{P} \sum_{a=1}^{P} \frac{\lambda_i}{\sqrt{\sum_{j=1}^{P} \lambda_j^2}} \frac{\mu_a}{\sqrt{\sum_{b=1}^{P} \mu_b^2}} M_{ia}. \tag{6}$$

Note that CKA is very similar to CCA but with additional (normalized) eigenvalue weighting. CKA will be the main focus of our work.

**Representational Similarity Analysis (RSA)** is a popular method in neuroscience used to compare different brain regions in response to the same set of stimuli [30]. It is similar to CKA, except RSA compares pair-wise Euclidean distances instead of pair-wise inner products. Recent work has established its equivalence to CKA when RSA is combined with an extra centering step [52]. Therefore, our analyses are directly applicable to (centered-)RSA.

## 3 Theoretical Background

Treating $\mathbf{\Sigma}_x$ and $\mathbf{\Sigma}_y$ as random matrices described in Sec. 2, we leverage results from random matrix theory [43] to compute deterministic equivalents of average CCA and CKA in the asymptotic limit. Defining $q_x = P/N_x$ and $q_y = P/N_y$, we consider the limit $P, N_x, N_y \to \infty$ by keeping $q_x, q_y \sim \mathcal{O}(1)$.

Both similarity measures depend on the cross-overlap between sample eigenvectors $M_{ia}$ defined in Eq. (4). Asymptotically $M_{ia}$ decouples as [7]

$$M_{ia} = \sum_{j,b} Q_{ij}^x \tilde{M}_{jb} Q_{ba}^y, \tag{7}$$

where the self-overlaps $Q_{ij}^x$ and $Q_{ab}^y$ can be computed analytically [32]. The self-overlap matrix for $\mathbf{X}$ can be expressed in terms of the resolvent matrix $\mathbf{G}(z) = (z - \mathbf{\Sigma})^{-1}$ given by:

$$Q_{ij} = C \lim_{\eta \to 0^+} \text{Im} \, \mathbf{G}_{jj}(\lambda_i - i\eta), \tag{8}$$

where $C$ is a constant and the resolvent $\mathbf{G}(z)$ has a deterministic equivalent defined by the following self-consistent equation

$$\mathbf{G}_{ij}(z) = \frac{\delta_{ij}}{z - \tilde{\lambda}_j(1 + q(z\mathfrak{g}(z) - 1))}, \quad \mathfrak{g}(z) = \frac{1}{P} \text{Tr} \, \mathbf{G}(z). \tag{9}$$

We provide a detailed derivation of these results in SI.A. Here, we note that the complex function $\mathfrak{g}(z)$ and Eq. (8) can be solved numerically (see SI.D for details).

**Main result.** Asymptotically, we obtain an analytical formula for sample CCA (Eq. (5)) and sample CKA (Eq. (6)) by replacing the cross-overlap matrix $M_{ia}$ with its deterministic equivalent (Eq. (7)).

Several remarks are in order:

– While the theory for CCA and CKA should generally apply to the cases where both models are sampled, henceforth, we fix one of the models to be deterministic for practical reasons. Often, neural similarity measures are applied to compare biological data with limited neuron recordings to an

---

[3]In the original SVCCA formulation [44], components are typically retained to explain a fixed proportion of variance. Our theoretical analysis applies regardless of the specific truncation criterion.

artificial model where the entire population is available. For example fixing model $\mathbf{Y}$ implies that its self-overlap $\mathbf{Q}^y$ is just an identity matrix, hence simplifying Eq. (7) to $\mathbf{M} = \mathbf{Q}^x\tilde{\mathbf{M}}$.

– The analytical formula for CCA and CKA depends only on the population quantities. However, since the self-overlap matrix $Q_{ij}$ in Eq. (8) explicitly depends on individual eigencomponents, its deterministic equivalent specifically depends on the expected sample eigenvalue for the $i^{\text{th}}$ component $(\mathbb{E}\lambda_i)$ and the population eigenvalue for the $j^{\text{th}}$ component $(\tilde{\lambda}_j)$. The latter makes it harder to apply the theory when the population eigenvalues cannot be observed. We discuss this issue further in Sec. 4.2.

**Sample Eigenvalues:** Theoretical values of individual sample eigenvalues $\mathbb{E}[\lambda_i]$ can be predicted given the population eigenvalues by solving the following integral equation [43]

$$\int_{\mathbb{E}[\lambda_i]}^{\infty} \rho(\lambda)\,d\lambda = \frac{i}{P}, \quad \rho(\lambda) = \frac{1}{\pi}\lim_{\eta\to 0^+}\text{Im}\,\mathfrak{g}(\lambda - i\eta), \tag{10}$$

where $\rho(\lambda)$ is the deterministic equivalent of the empirical eigenvalue density (see SI.A.2). Computing $\mathbb{E}[\lambda_i]$ this way may be problematic due to numerical instabilities. Alternatively, one can exploit the fact that each single-trial eigenvalue concentrates around this mean with trial-to-trial fluctuations of $\mathcal{O}(1/\sqrt{P})$ [43] and simply replace $\mathbb{E}[\lambda_i]$ with a single-trial observation in the large $P$ limit. We provide a detailed account of this approximation in SI.A.5.

**Sample Eigenvectors:** Unlike eigenvalues, the sample eigenvectors $\langle u_i|\tilde{u}_j\rangle^2$ exhibit trial-to-trial fluctuations that persist even as $P \to \infty$ (see SI.A.6). Still, we can compute the mean value of the overlap represented by the squared overlap $Q_{ij}$ in Eq. (3).

**BBP Phase Transition:** In addition to inevitable fluctuation in the sample eigenvectors, their mean behavior can still differ markedly from that of the population eigenvectors. A classic example is the Baik–Ben Arous–Péché (BBP) phase transition [4]. Consider a population Gram matrix with one large "spike" eigenvalue and the rest equal to 1. Depending on whether the spike strength exceeds a critical threshold determined by $P/N$, the sample eigenvector associated with it can either have an $\mathcal{O}_P(1)$ with the true eigenvector (localized) or can be completely uncorrelated (delocalized). We depict this transition in Fig. 1a.

**Numerical Confirmation:** Finally, we numerically test the theoretical prediction for self-overlap given by Eq. (8) on the eigenvectors of deep neural network activations. We extract layer activations from a pre-trained ResNet18 on CIFAR-10 images and subsample $N$ neurons through random projection. In Fig. 1b, we show the self-overlap $Q_{ii}$ for the first few eigenvectors of the layer activations and demonstrate a perfect match with theory. As the number of neurons decreases, the number of delocalized eigenvectors increases since fewer eigenvectors have self-overlap $Q_{ii} \approx 1$.

The effect of eigenvector delocalization[4] is reflected in the CKA between the sampled and population layer activations as shown in Fig. 1c. The alignment is completely misleading when small numbers of neurons are sampled, which poses a significant problem for practical purposes.

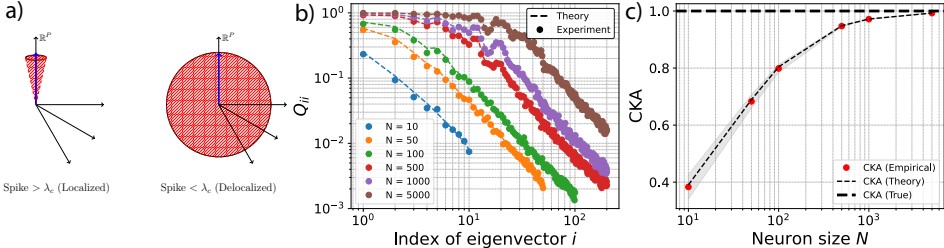

Figure 1: **a)** Illustration of eigenvector delocalization in BBP phase transition. **b)** Self-overlap $Q_{ii}$ between sample and population eigenvectors for ResNet18 activations. **c)** CKA between population and sample activations when $N$ neurons are sampled. The gray-shaded region represents the standard deviation of empirical CKA across different random samplings.

---

[4]While eigenvalues also change, we show in SI.F that the dominant factor in reduced CKA is the eigenvector delocalization, especially for fast decaying spectra.

# 4    Applying Theory to Representation Similarity

## 4.1    Forward Problem: Impact of Neuron Sampling on Similarity

In the forward problem, we assume that the population eigenvalues and eigenvectors are known. The first step is to obtain the typical sample eigenvalues by running a single-trial numerical simulation. We then move on to the eigenvectors by computing $\mathbf{Q}$ using Eq. (8). Finally, we calculate the overlap between the two systems, $\mathbf{M}$, using Eq. (7). Having these components allows us to evaluate both CCA and CKA as functions of the number of neurons $N$.

As illustrated in Fig. 2, the theoretical predictions obtained from this eigen-decomposition match the observed CCA and CKA across different values of $N$. Notice that CKA decreases when the number of neurons $N$ is reduced. As discussed above, both of these effects can be explained by the delocalization of eigenvectors.

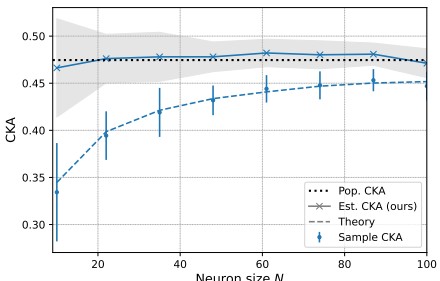 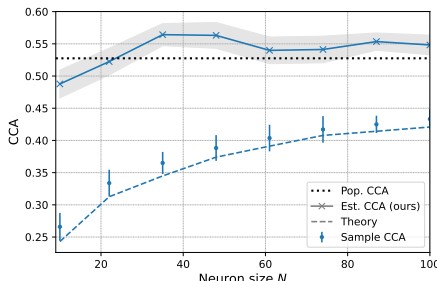

Figure 2: **Comparison of sample vs population measures for CKA and CCA:** Error bars represent empirical sample similarity and dotted lines the theoretical predictions. The black dotted line marks the true population similarity which is set close to 0.5 for both measures. Solid lines indicate inferred true similarity from samples. Sample similarity is lower due to eigenvector delocalization, while our method consistently provides a closer estimate of the true value.

## 4.2    Backward Problem: Inferring Population Similarity from Limited Neurons

Just like in our earlier analysis, inferring the population representational similarity begins with estimating the eigenvalues of the underlying population. In general, this is difficult because sample eigenvalues can deviate substantially from their population counterparts. Moreover, if $N < P$, there are $P - N$ zero eigenvalues in the sample covariance matrix, further complicating the problem.

However, if we adopt a parametric form, we can often achieve significant improvements in accuracy [41]. Here, we assume a power-law spectrum of the form $\tilde{\lambda}_i = i^{-1-\gamma}$, and develop a numerical method based on random matrix theory that reliably infers the true decay rate of population eigenvalues based on only the sample eigenvalues (see SI.C for detailed analysis). One can also consider more sophisticated eigenvalue models (e.g. broken power-law [41]) are also possible.

While it is possible to estimate the population eigenvalues for general spectra [33], they require estimating each eigenvalue individually and are hence computationally expensive. Here, we only consider the power-law spectrum because 1) we only need to estimate a single parameter and 2) we can derive a closed-form expression for the population eigenvalues (see SI.C for derivation) and 3) it has been shown to be relevant for biological systems [49].

After estimating the population eigenvalues $\{\tilde{\lambda}_i\}$, we address the eigenvectors by computing the self-overlap matrix $\mathbf{Q}$ using Eq. (8). Since every population eigenbasis produces the same mean self-overlap, estimated population eigenvalues $\{\tilde{\lambda}_i\}$ are sufficient to find $\mathbf{Q}$.

Our final goal is to estimate the population cross-overlaps $\tilde{\mathbf{M}}$, which are required to infer the true population similarity between two systems. Here, we propose a constrained optimization problem to invert the forward relationship $\mathbf{M} = \mathbf{Q}\tilde{\mathbf{M}}$ using the estimated $\mathbf{Q}$ and the observed $\mathbf{M}$, as shown in Alg. 1.

Two challenges arise in this naive approach. First, eigenvector statistics do not self-average [43], so the empirical cross-overlap $\mathbf{M}$ deviates from its expected value. This discrepancy can be partially mitigated by trial averaging or statistical bootstrapping. Second, the self-overlap $\hat{\mathbf{Q}}$ is not invertible unless $P \ll N$. As a result, it is impossible to recover the entire matrix $\tilde{\mathbf{M}}$. Intuitively, only the first few eigenvectors are well-localized; the rest delocalize and lose information, so we can only reliably retrieve the corresponding columns of $\tilde{\mathbf{M}}$.

While the constrained optimization provides point estimates of $\tilde{\mathbf{M}}$, it does not directly quantify their uncertainty. To assess the reliability of these estimates, we additionally derive confidence intervals under a maximum likelihood estimation (MLE) framework. This allows us to estimate statistical uncertainty for each $\tilde{M}_{ja}$, as well as for composite quantities such as CKA and CCA (see SI.G).

---

**Algorithm 1** Inferring Pop. Cross-Overlap $\tilde{\mathbf{M}}$

**Require:** $\{\lambda_i\}_{i=1}^P$: Sample eigenvalues
1:      $P$: # of stimuli , $N$: # of neurons
2:      $\mathbf{M} \in \mathbb{R}^{P \times P}$: sample cross-overlap
3: **Step 1: Estimate Population Eigenvalues**
4:      Assume power-law ansatz: $\tilde{\lambda}_i \propto i^{-1-\gamma}$
5:      Find $\gamma$ that best explains $\{\lambda_i\}_{i=1}^P$

6: **Step 2: Compute Self-overlap matrix**
7:      $\mathbf{Q} \leftarrow function(\{\tilde{\lambda}_i\}, P, N)$
8: **Step 3: Optimize Population Similarity**
9:      $\tilde{\mathbf{M}}_{est} \leftarrow \arg\min_{\tilde{\mathbf{M}}} \|\mathbf{M} - \mathbf{Q} \cdot \tilde{\mathbf{M}}\|_F$
10:          s.t. $\tilde{M}_{ij} \in [0,1]$ for $\forall i, j$
    **return** $\tilde{\mathbf{M}}_{est}$

---

#### 4.2.1 Up to How Many Eigenvectors Can We Resolve for Given $N, P$?

Consider a power-law spectrum, which decays relatively quickly. Under such a spectrum, only the leading sample eigenvectors tend to be well-localized, as shown in Fig. 3(Left). If we run the backward algorithm, we observe that for a given $N, P$, we can reliably recover only those initial components that remain localized, as shown in Fig. 3(Right).

**Practical implication.** For power-law population spectra $\tilde{\lambda}_i \propto i^{-1-\gamma}$, the critical localization index scales as $i^\star(N) \approx \frac{1+\gamma}{\sqrt{8}}\sqrt{N}$ (see SI.H). This provides a simple rule of thumb: with $N$ recorded neurons, one can reliably resolve roughly $\frac{1+\gamma}{\sqrt{8}}\sqrt{N}$ leading eigenvectors. Conversely, to stably recover the top $k$ components, it is sufficient to record $N \gtrsim \frac{8}{(1+\gamma)^2}k^2$ neurons.

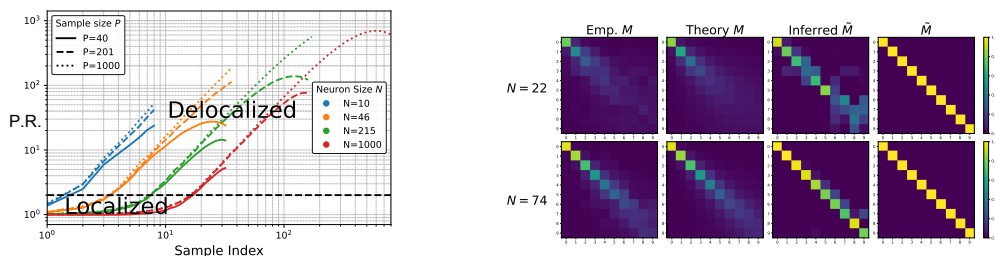

Figure 3: **Left:** Participation ratio (P.R.) of self-overlap ($1/\sum_j Q_{ij}^2$), indicating the onset of eigenvector delocalization, for a power-law spectrum $\tilde{\lambda}_i \sim i^{-1.2}$. For fixed $N$, increasing $P$ marginally affects the leading eigenvectors. By contrast, for fixed $P$, increasing $N$ makes more eigenvectors localized. Only sample eigenvectors below the black horizontal line are localized (P.R. $\approx 1$). Heuristically, $\tilde{M}_{ia}$ can be recovered reliably for only indices below this line. **Right:** Each column shows the 5-trial averaged $\mathbf{M}$, the theoretical prediction of $\mathbf{M}$, the inferred population overlap $\tilde{\mathbf{M}}_{est}$, and the actual population overlap $\tilde{\mathbf{M}}$. With fewer neurons $N$, sample eigenvectors become delocalized, causing large discrepancies. Nevertheless, our inference method successfully recovers the dominant overlaps, which are enough for global similarity measures such as CKA and CCA.

We can explicitly truncate these eigenvectors by taking a partial inverse of $\mathbf{Q}$ (see SI.E). However, this approach can be numerically unstable and might produce values of $M_{ij}$ outside the $[0,1]$ range.

Additionally, Fig. 3(Left) demonstrates that, under a power-law of the same exponent, varying $P$ has a subtler effect on these leading indices than varying $N$, which significantly affects localization.

### 4.2.2 Why This Is Sufficient for Inferring Population Similarity

Although our denoising approach only manages to recover the leading few eigencomponents (those that remain localized), it is precisely these components that matter most for similarity measures like CKA and (SV)CCA. As shown in Fig. 2, these metrics are governed primarily by the initial eigenvalues and eigenvectors. Thus, even with a very limited number of neurons, estimating those leading components is sufficient for practical purposes.

Note that for CKA (and not CCA), there is an alternative approach to infer population similarity called the moment-based estimator [22, 28, 14], which computes the similarity using unbiased statistics. This method is more suitable for estimating similarity with small datasets but lacks theoretical insight. In contrast, our approach provides an analytical framework for studying how spectral properties precisely alter the observed similarity, but it assumes sufficiently large datasets so that all biases are negligible.

## 5 Experiments

### 5.1 Synthetic Data with a Known Population Gram Matrix

We first evaluate our approach on a synthetic dataset where the population Gram matrix is fully specified, allowing us to directly compare our estimated similarity measures against the ground-truth population values.

Fig. 2 illustrates that our forward and backward procedures work well. In the forward approach, we show that the eigencomponent-based analysis matches the empirical results closely. In the backward approach, even with an extremely limited number of neurons ($N \approx 20$), our method infers a population similarity close to the actual value, despite the observed sample similarity being substantially lower.

Since the population eigenvectors are known, we can also verify how well the inferred overlaps match the true overlaps. Specifically, Fig. 3(Right) displays the top-left $10 \times 10$ block of each matrix: the empirical $\mathbf{M}$, the theoretical $\mathbf{M}$ (second column), the inferred population overlap $\tilde{\mathbf{M}}_{est}$ (third column), and the actual population overlap $\tilde{\mathbf{M}}$ (fourth column). In this example, we set $\tilde{\mathbf{\Sigma}}_x = \tilde{\mathbf{\Sigma}}_y$, and hence the actual population cross-overlap should be the identity matrix. However, with fewer neurons, the sample eigenvectors become more delocalized, as evident in the first column. The theoretical prediction of this phenomenon (second column) aligns closely with the empirical observation. Notably, even with severely limited neurons, our backward-inference method recovers a cross-overlap matrix $\tilde{\mathbf{M}}_{est}$ (third column) much closer to the true identity than the naive observed $\mathbf{M}$.

### 5.1.1 Sampling Neurons Can Change Representation Similarity Ranking

Next, we showcase a synthetic example in which *sampling* can lead to a reversal in the similarity rankings of models. Specifically, we construct two models:

- **Model 1** has significant overlap with the Brain on its first 3 population eigenvectors.
- **Model 2** has significant overlap with the Brain on the next 3 eigenvectors.

We set the total population (SV)CCA of Model 2 to be higher than that of Model 1. However, as neurons are sampled, eigenvectors corresponding to larger indices (smaller eigenvalues) tend to delocalize more. Hence, the empirical cross-overlap $\mathbf{M}$ for Model 2 deteriorates faster, causing its (SV)CCA to drop more than that of Model 1. Eventually, Model 1 overtakes Model 2 in the sample-based (SV)CCA ranking, as illustrated in Fig. 4(Left).

Fig. 4(Right) presents the empirical and population cross-overlaps of the two models (each compared to the Brain). We set $P = 200$ and $N = 30$, and all population eigenvalues follow a power-law with exponent $-1.2$. Model 2's higher-dimensional overlaps delocalize more strongly, producing an apparent discrepancy that flips their observed ranking once neuron sampling is taken into account.

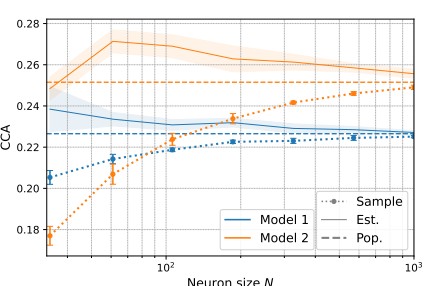


Figure 4: **Left:** Sample-based CCA ranking flips despite Model 2 having a larger *population* CCA than Model 1. The decrease in Model 2's CCA is more pronounced due to its stronger reliance on higher-indexed eigenvectors, which become more delocalized with limited neuron sampling. **Right:** Empirical vs. population cross-overlaps for Model 1 vs. Brain and Model 2 vs. Brain. Here, $P = 200$ and $N = 30$. All three population eigenvalue spectra follow a power-law with exponent $-1.2$. Although Model 2's true overlap is higher at the population level, it relies on higher-indexed (smaller eigenvalue) components, which delocalize more severely in the sample.

## 5.2 Brain Data

Finally, we apply our denoising framework to real neural recordings in the primate visual cortex, comparing them against various computational model predictions. (for experimental details see SI.D)

In Fig. 5, we illustrate a scatter plot of the representation similarity for different models compared to neural responses from V2 cortex [18, 46], given an artificially limited neuron count of $N = 20$ out of 103 neurons. The $x$-axis corresponds to the observed *sample* CKA or CCA, while the $y$-axis is our *inferred population* measure. Observe that our inference method consistently produces higher population similarity estimates than the naive sample estimates. In particular, certain models that appear to have lower similarity (when judged by the raw, sample-based metric) can actually exhibit higher *true* similarity to the brain once sampling effects are taken into account.

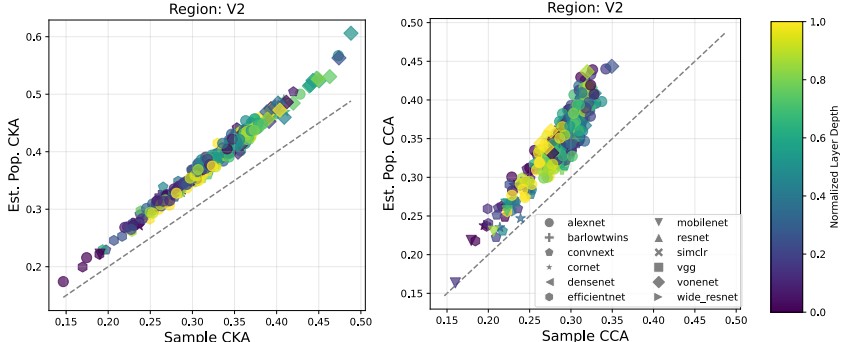

Figure 5: Scatter plots of observed sample similarity vs. inferred population similarity for multiple models compared to V2 cortex, using only $N = 20$ neurons (out of a larger set). (**Left**) CKA results; (**Right**) CCA results. The dotted line $y = x$ indicates equality. Notice that the inferred population similarity is consistently higher than the naive sample-based measure, demonstrating how limited neuron sampling can lead to underestimation of the true model-brain correspondence.

## 6 Conclusion and Outlook

We have presented an eigencomponent-based analysis of how sampling a finite number of neurons affects similarity measures, including CCA and CKA. By applying methods from Random Matrix Theory, we established that this limited sampling systematically underestimates similarity because of eigenvector delocalization in the sample Gram matrices. Our framework provides:

- **Forward Analysis:** Predicting how *Population* eigenvalues and eigenvectors will manifest under neuron sampling, thus explaining the observed drop in similarity.
- **Backward Inference:** Denoising algorithm to infer *Population* representation similarity from limited data, overcoming the biases introduced by sampling noise.

We validated our approach on both synthetic and real datasets. In the synthetic experiments, where the population Gram matrices were fully known, we showed that our method reliably recovers the true population overlaps and similarity values, even in regimes with very few neurons. Importantly, we highlighted a striking effect of sampling: under certain configurations, the ranking of two models with respect to the brain can be inverted when only a limited set of neurons is recorded. In real datasets from primate visual cortex, our method consistently produced higher *population* similarity estimates than naive sample-based methods, underscoring that the observed decrease in similarity is largely a sampling artifact.

Moreover, for representations with power-law eigenspectra, we identified a universal scaling law: the number of well-localized eigenvectors grows as the square root of the number of recorded neurons. This $\sqrt{N}$ behavior offers a practical rule of thumb—researchers can estimate how many principal components can be reliably resolved for a given neuron count, or conversely, how many neurons are needed to capture a desired number of components (see SI.H). This scaling bridges theoretical predictions with experimental design, guiding how to interpret and plan neural recording studies under finite sampling constraints.

**Future Directions.**   There are several promising avenues for extending our work. First, it would be valuable to explore more sophisticated spectral priors—such as broken power-law spectra—to account for multiple functional subpopulations in the data, each contributing a distinct spectral structure. Second, while we have focused on sampling noise, future work should incorporate explicit models of additive noise that arise in real-time neurophysiological recordings, relaxing the assumption that trial averaging eliminates most of it. Third, improved denoising methods could be developed by adopting Bayesian approaches to model the joint distribution of sample eigenvectors and population eigenvectors [38], thus allowing more accurate recovery of the population eigenspaces. Finally, as we outline in SI.B, our framework naturally extends to regression settings, where sampling-induced distortions in eigencomponents can adversely affect regression scores, much like their impact on representational similarity measures.

Overall, our results suggest that practical neuroscience studies must account for sampling-induced eigenvector delocalization when interpreting representational similarity. By unveiling the intrinsic biases introduced by limited neuron sampling and proposing a systematic solution, we aim to provide neuroscientists and machine learning researchers with more reliable tools for comparing computational models and neural data.

# 7   Limitations

Our framework assumes that neural responses arise from Gaussian (linear) projections of latent population codes. This yields analytical tractability via random matrix theory, but real data can exhibit non-Gaussian statistics, nonlinearities, and stimulus-dependent covariances, which may reduce the quantitative accuracy of our estimators when higher-order dependencies dominate.

In addition, the similarity measures we study rely on specific symmetry assumptions; our method currently models rotational invariance only, ignoring other relevant symmetries (e.g., translation, scaling, permutation). It also does not yet cover dynamic (time-resolved/trajectory) or nonlinear similarity metrics. Extending the theory to these richer classes remains a crucial area for future research.

## Acknowledgments and Disclosure of Funding

This work was supported by the Center for Computational Neuroscience at the Flatiron Institute of the Simons Foundation, as well as by a Sloan Research Fellowship and a Klingenstein-Simons Award (to S.C.). All experiments were performed on the high-performance computing cluster at the Flatiron Institute.

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

# A  Detailed Derivation of the Main Result

## A.1  The Sample Gram Matrix

Let $\tilde{\mathbf{X}} \in \mathbb{R}^{P \times \tilde{N}_x}$ denote the true population matrix with $P$ samples and $\tilde{N}_x$ neurons. We consider sampling only in the neuron/feature axis. The sample data $\mathbf{X} \in \mathbb{R}^{P \times N_x}$ is obtained by applying an $\tilde{N}_x \times N_x$ random projection matrix $\mathbf{R}_x$ on $\tilde{\mathbf{X}}$

$$\mathbf{X} = \tilde{\mathbf{X}}\mathbf{R}_x, \quad (\mathbf{R}_x)_{ij} \sim \mathcal{N}\left(0, \frac{1}{N_x}\right). \tag{S1}$$

The population and sample Gram matrices and their corresponding eigencomponents are denoted as

$$\tilde{\boldsymbol{\Sigma}}_x = \tilde{\mathbf{X}}\tilde{\mathbf{X}}^\top = \sum_{i=1}^{P} \tilde{\lambda}_i \left|\tilde{u}_i\right\rangle\!\left\langle\tilde{u}_i\right|,$$

$$\boldsymbol{\Sigma}_x = \mathbf{X}\mathbf{X}^\top = \sum_{i=1}^{P} \lambda_i \left|u_i\right\rangle\!\left\langle u_i\right|. \tag{S2}$$

In Random Matrix Theory (RMT), it is often convenient to consider matrices of the form $\mathbf{M} = \sqrt{\mathbf{C}}\mathbf{W}\sqrt{\mathbf{C}}$, where $\mathbf{W} = \mathbf{R}\mathbf{R}^\top$ is a random Wishart matrix and $\mathbf{C}$ is a deterministic square matrix. We first put $\boldsymbol{\Sigma}_x$ into this form to simplify our calculations [27]. The sample Gram matrix can be written in terms of the SVD components of $\tilde{\mathbf{X}} = \mathbf{U}\tilde{\boldsymbol{\Lambda}}_x^{1/2}\mathbf{V}^\top$

$$\boldsymbol{\Sigma}_x = \tilde{\mathbf{X}}\mathbf{R}_x\mathbf{R}_x^\top\tilde{\mathbf{X}}^\top = \mathbf{U}\tilde{\boldsymbol{\Lambda}}_x^{1/2}\left(\mathbf{V}^\top\mathbf{R}_x\mathbf{R}_x^\top\mathbf{V}\right)\tilde{\boldsymbol{\Lambda}}_x^{1/2}\mathbf{U}^\top, \tag{S3}$$

where $\tilde{\boldsymbol{\Lambda}}_x \in \mathbb{R}^{P \times \tilde{N}_x}$ is a diagonal matrix, and $U \in \mathbb{R}^{P \times P}$ and $V \in \mathbb{R}^{\tilde{N}_x \times \tilde{N}_x}$ orthogonal matrices. Since deterministic orthogonal transformations of Wishart matrices are again Wishart matrices, we get:

$$\boldsymbol{\Sigma}_x = \mathbf{U}\tilde{\boldsymbol{\Lambda}}_x^{1/2}\mathbf{W}_x\tilde{\boldsymbol{\Lambda}}_x^{1/2}\mathbf{U}^\top, \tag{S4}$$

where $\mathbf{W}_x = \mathbf{V}^\top\mathbf{R}_x\mathbf{R}_x^\top\mathbf{V}$ is a random Wishart matrix with aspect ratio $\phi_x = \tilde{N}_x/N_x$. We divide our discussion into two cases:

- When $P \geq \tilde{N}_x$, the eigenvalue matrix can be completed to a $P \times P$-matrix by zero padding and replacing $\mathbf{W}_x$ with a Wishart matrix with $q_x = P/N_x$. Using the orthogonality of $\mathbf{U}$, this allows us to express $\boldsymbol{\Sigma}_x$ as

$$\boldsymbol{\Sigma}_x = (\mathbf{U}\tilde{\boldsymbol{\Lambda}}_x^{1/2}\mathbf{U}^\top)(\mathbf{U}\mathbf{W}_x\mathbf{U}^\top)(\mathbf{U}\tilde{\boldsymbol{\Lambda}}_x^{1/2}\mathbf{U}^\top) = \sqrt{\tilde{\boldsymbol{\Sigma}}_x}\mathbf{W}_x\sqrt{\tilde{\boldsymbol{\Sigma}}_x}, \tag{S5}$$

  where $\mathbf{W}_x$ is a Wishart matrix with aspect ratio $q_x = P/N_x$.

- When $P < N_x$, the eigenvalue matrix and the Wishart matrix can be written as

$$\tilde{\boldsymbol{\Lambda}}_x = \begin{pmatrix}\tilde{\boldsymbol{\Lambda}}_x' & 0\end{pmatrix}, \quad \mathbf{W}_x = \begin{pmatrix}\mathbf{R}_1 \\ \mathbf{R}_2\end{pmatrix}\begin{pmatrix}\mathbf{R}_1^\top & \mathbf{R}_2^\top\end{pmatrix}, \tag{S6}$$

  where the $P \times P$ matrix $\tilde{\boldsymbol{\Lambda}}_x'$ is the non-zero part of $\tilde{\boldsymbol{\Lambda}}_x$ and $\mathbf{R}_1 \in \mathbb{R}^{P \times N_x}$, $\mathbf{R}_2 \in \mathbb{R}^{(\tilde{N}_x - P) \times N_x}$ are two projection matrices. Plugging these back in, we arrive at the same form as the previous case.

In both cases, the statistics of $\boldsymbol{\Sigma}_x$ does not depend explicitly on $\tilde{N}_x$.

## A.2  Eigenvalue statistics of sample Gram matrices

One of the main objectives of RMT is to understand the eigenvalue distribution of random matrices in terms of deterministic quantities [43]. Here, we review some classical results on the eigenvalue statistics of random matrices of the form $\boldsymbol{\Sigma} = \sqrt{\tilde{\boldsymbol{\Sigma}}}\mathbf{W}\sqrt{\tilde{\boldsymbol{\Sigma}}}$ where $\mathbf{W}$ is a $P \times N$ Wishart matrix

with ratio $q = \frac{P}{N}$. Here, $\boldsymbol{\Sigma}$ and $\tilde{\boldsymbol{\Sigma}}$ are the sample and population Gram matrices, and they have the following eigendecompositions

$$\boldsymbol{\Sigma} = \sum_{i=1}^{P} \lambda_i \left| u_i \right\rangle\!\left\langle u_i \right|, \qquad \tilde{\boldsymbol{\Sigma}} = \sum_{i=1}^{P} \tilde{\lambda}_i \left| \tilde{u}_i \right\rangle\!\left\langle \tilde{u}_i \right|. \tag{S7}$$

We denote their (discrete-)eigenvalue distribution by $\rho(\lambda)$ and $\tilde{\rho}(\tilde{\lambda})$:

$$\rho(\lambda) = \frac{1}{P} \sum_{i=1}^{P} \delta(\lambda - \lambda_i), \qquad \tilde{\rho}(\tilde{\lambda}) = \frac{1}{P} \sum_{i=1}^{P} \delta(\tilde{\lambda} - \tilde{\lambda}_i). \tag{S8}$$

We define the resolvent of the random matrix $\mathbf{X}$ and its trace as

$$\mathbf{G}(z) = (z - \boldsymbol{\Sigma})^{-1} = \sum_{i=1}^{P} \frac{\left| u_i \right\rangle\!\left\langle u_i \right|}{z - \lambda_i}. \tag{S9}$$

The Stieltjes transform of the empirical spectral distribution is defined as

$$\mathfrak{g}^P(z) := \int \frac{\rho(\lambda)}{z - \lambda} d\lambda = \frac{1}{P} \operatorname{Tr} \mathbf{G}(z). \tag{S10}$$

In the large $P$ limit, this quantity is self-averaging and there is a deterministic equivalent $\mathfrak{g}(z) \sim \mathfrak{g}^P(z)$ given by the self-consistent equation

$$\mathfrak{g}(z) = \int \frac{\tilde{\rho}(\tilde{\lambda})}{z - \tilde{\lambda}(1 - q + qz\mathfrak{g}(z))} d\tilde{\lambda}, \tag{S11}$$

which only depends on the deterministic eigenvalues $\tilde{\rho}_x(\tilde{\lambda})$ and the ratio $q = P/N$ [43]. In practical applications, $\tilde{\rho}_x(\tilde{\lambda})$ is often replaced with the uniform measure over the population eigenvalues $\{\tilde{\lambda}_i\}$ as defined in Eq. (S8). This remarkable result was first obtained in [37] for white Wishart matrices (for which $\tilde{\boldsymbol{\Sigma}} = \mathbf{I}$).

Due to the equivalence $\mathfrak{g}(z) \sim \mathfrak{g}^P(z)$ in large $P$ limit, these two integrals are equivalent

$$\int \frac{\rho(\lambda)}{z - \lambda} d\lambda \underset{P \to \infty}{\to} \int \frac{\tilde{\rho}(\tilde{\lambda})}{z - \tilde{\lambda}(1 - q + qz\mathfrak{g}(z))} d\tilde{\lambda}, \tag{S12}$$

from which one can obtain the density of the limiting spectral density using the inversion formula [8]

$$\rho(\lambda) = \frac{1}{\pi} \lim_{\eta \to 0^+} \operatorname{Im} \mathfrak{g}(\lambda - i\eta). \tag{S13}$$

The Stieltjes transform also connects to the effective regularization in ridge regression [6, 24, 11, 2]. We define a new function $\kappa(z)$ as

$$\kappa(z) := -\frac{z}{1 - q + qz\mathfrak{g}(z)}, \quad \mathfrak{g}(z) = z^{-1} - q^{-1}(z^{-1} + \kappa(z)^{-1}) \tag{S14}$$

and express Eq. (S11) in terms of this quantity:

$$\mathfrak{g}(z) = \frac{\kappa(z)}{z} \int \frac{\tilde{\rho}(\tilde{\lambda})}{\tilde{\lambda} + \kappa(z)} d\tilde{\lambda} = z^{-1} - q^{-1}(z^{-1} + \kappa(z)^{-1}). \tag{S15}$$

Then, we obtain a new self-consistent equation for $\kappa$

$$\kappa(z) = -z + \kappa(z) \int \frac{q\tilde{\lambda}}{\tilde{\lambda} + \kappa(z)} \tilde{\rho}(\tilde{\lambda}) d\tilde{\lambda}, \tag{S16}$$

which is also known as the Silverstein equation [48]. Expressing this in terms of the discrete population eigenvalues, and evaluating it at $z = -\lambda$, we get

$$\kappa = \lambda + \kappa \frac{1}{N} \sum_{i=1}^{P} \frac{\tilde{\lambda}_i}{\tilde{\lambda}_i + \kappa}, \tag{S17}$$

which is exactly the equation for the renormalized ridge parameter in [11, 2] with the scaling $\tilde{\lambda}_i \to N\tilde{\lambda}_i$.

## A.3 Eigenvector statistics of sample Gram matrices and the self-overlap matrix

This result from Eq. (S11) can also be generalized to the resolvent matrix itself [27, 8], which becomes diagonal in the population eigenbasis:

$$\mathbf{G}(z) = \sum_{i=1}^{P} \frac{|u_i\rangle\langle u_i|}{z - \lambda_i} \sim \sum_{i=1}^{P} \frac{|\tilde{u}_i\rangle\langle\tilde{u}_i|}{z - \tilde{\lambda}_i(1 - q + qz\mathfrak{g}(z))}, \tag{S18}$$

where the integral over eigenvalues is replaced by the discrete measure over population eigenvalues. This allows us to study the eigenvector statistics by analyzing the quantity

$$\langle\tilde{u}_j|\mathbf{G}(z)|\tilde{u}_j\rangle = \sum_{i=1}^{P} \frac{\langle u_i|\tilde{u}_j\rangle^2}{z - \lambda_i} \sim \frac{1}{z - \tilde{\lambda}_j(1 - q + qz\mathfrak{g}(z))}. \tag{S19}$$

In the large $P$ limit, the sum over empirical eigenvalues becomes an integral:

$$\langle\tilde{u}_j|\mathbf{G}(z)|\tilde{u}_j\rangle \xrightarrow[P\to\infty]{} \int \frac{Q(\lambda, \tilde{\lambda}_j)}{z - \lambda}\rho(\lambda)d\lambda, \tag{S20}$$

where we defined $Q(\lambda_i, \tilde{\lambda}_j) := P\langle u_i|\tilde{u}_j\rangle^2$ is the overlap between the $i^{\text{th}}$ sample eigenvector and the $j^{\text{th}}$ population eigenvector. Now, we can obtain $Q(\lambda_i, \tilde{\lambda}_j)$ using the following inversion formula

$$Q(\lambda_i, \tilde{\lambda}_j) = \frac{1}{\pi\rho(\lambda_i)} \lim_{\eta\to 0^+} \text{Im}\,\langle\tilde{u}_j|\mathbf{G}(\lambda_i - i\eta)|\tilde{u}_j\rangle. \tag{S21}$$

Using the equivalence in Eq. (S19) and evaluating this expression explicitly:

$$Q(\lambda_i, \tilde{\lambda}_j) = \frac{q\lambda_i\tilde{\lambda}_j}{\left[\tilde{\lambda}_j(1 - q) - \lambda_i + q\lambda_i\tilde{\lambda}_j\mathfrak{h}(\lambda_i)\right]^2 + \left[q\lambda_i\tilde{\lambda}_j\pi\rho(\lambda_j)\right]^2}, \tag{S22}$$

we get an explicit formula for eigenvector overlaps [32, 8], where $\rho(\lambda_i)$ is given by Eq. (S13) and $\mathfrak{h}(z)$ is its Hilbert transform:

$$\mathfrak{h}(z) = \text{p.v.} \int \frac{\rho(\lambda)}{z - \lambda}d\lambda. \tag{S23}$$

and can be obtained from the Stieltjes transform via

$$\lim_{\eta\to 0^+} \mathfrak{g}(z - i\eta) = \mathfrak{h}(z) + i\pi\rho(z). \tag{S24}$$

## A.4 Overlap formula for two Gram matrices

Here, we provide a short review of the work by Bun et al. [7] which derives an overlap formula between eigenvectors from random matrices. We consider observations from two representations $\mathbf{X} \in \mathbb{R}^{P \times N_x}$ and $\mathbf{Y} \in \mathbb{R}^{P \times N_y}$ in response to a common set of inputs of size $P$. Their sample Gram matrices have decompositions:

$$\mathbf{\Sigma}_x = \mathbf{X}\mathbf{X}^\top = \sum_{i=1}^{P} \lambda_i |u_i\rangle\langle u_i|, \qquad \mathbf{\Sigma}_y = \mathbf{Y}\mathbf{Y}^\top = \sum_{a=1}^{P} \mu_a |w_a\rangle\langle w_a|. \tag{S25}$$

We assume that $\mathbf{X}$ and $\mathbf{Y}$ are observations sampled from the underlying population features $\tilde{\mathbf{X}} \in \mathbb{R}^{P \times \tilde{N}_x}$ and $\tilde{\mathbf{Y}} \in \mathbb{R}^{P \times \tilde{N}_y}$ through independent random projections. The corresponding population Gram matrices are decomposed as:

$$\tilde{\mathbf{\Sigma}}_x = \tilde{\mathbf{X}}\tilde{\mathbf{X}}^\top = \sum_{i=1}^{P} \tilde{\lambda}_i |\tilde{u}_i\rangle\langle\tilde{u}_i|, \qquad \tilde{\mathbf{\Sigma}}_y = \tilde{\mathbf{Y}}\tilde{\mathbf{Y}}^\top = \sum_{a=1}^{P} \tilde{\mu}_a |\tilde{w}_a\rangle\langle\tilde{w}_a|. \tag{S26}$$

We consider two sample data matrices $\mathbf{X} \in \mathbb{R}^{P \times N_x}$ and $\mathbf{Y} \in \mathbb{R}^{P \times N_y}$. In Sec. A.1, we showed that the sample Gram matrices can be expressed in terms of the population ones as:

$$\mathbf{\Sigma}_x = \sqrt{\tilde{\mathbf{\Sigma}}_x} \mathbf{W}_x \sqrt{\tilde{\mathbf{\Sigma}}_x},$$
$$\mathbf{\Sigma}_y = \sqrt{\tilde{\mathbf{\Sigma}}_y} \mathbf{W}_y \sqrt{\tilde{\mathbf{\Sigma}}_y}, \tag{S27}$$

where the Wishart matrices $\mathbf{W}_x$ and $\mathbf{W}_y$ have aspect ratios $q_x = P/N_x$ and $q_y = P/N_y$, respectively. Resolvents of the sample Gram matrices are

$$\mathbf{G}_x(z) \equiv (z - \mathbf{\Sigma}_x)^{-1} = \sum_{i=1}^{P} \frac{|u_i\rangle\langle u_i|}{z - \lambda_i}, \qquad \mathbf{G}_y(z') \equiv (z' - \mathbf{\Sigma}_y)^{-1} = \sum_{a=1}^{P} \frac{|w_a\rangle\langle w_a|}{z' - \mu_a}. \tag{S28}$$

We want to compute

$$\psi_P(z, z') = \mathbb{E}\left[\frac{1}{P} \operatorname{Tr}\left[\mathbf{G}_x(z)\mathbf{G}_y(z')\right]\right] = \mathbb{E}\left[\frac{1}{P^2} \sum_{i,a=1}^{P} \frac{P \langle u_i | w_a \rangle^2}{(z - \lambda_i)(z' - \mu_a)}\right], \tag{S29}$$

where the expectation is over random realizations of sample Gram matrices [7]. In the limit $P \to \infty$, as empirical eigenvalues become continuous, this object approaches a deterministic function

$$\psi_P(z, z') \sim \psi(z, z') = \int \frac{\rho_x(\lambda)\rho_y(\mu)}{(z - \lambda)(z' - \mu)} M(\lambda, \mu) d\lambda \, d\mu, \quad M(\lambda_i, \mu_a) \sim \mathbb{E}\left[P \langle u_i | w_a \rangle^2\right] \tag{S30}$$

Here, $\rho_x(\lambda)$, $\rho_y(\mu)$ are the eigenvalue densities of $\mathbf{\Sigma}_x$, $\mathbf{\Sigma}_y$ given by Eq. (S13). The function $M(\lambda_i, \mu_a) \sim \mathbb{E}\left[P \langle u_i | w_a \rangle^2\right]$ denotes the expected overlap between two eigenvectors associated with eigenvalues $\lambda_i$ and $\mu_a$, and it is the central object for our analysis since it directly appears in CCA and CKA. This quantity can be obtained by computing $\psi(\lambda_i - i\eta, \mu_a + i\eta')$, collecting the term proportional to $\eta\eta'$ and taking the limit $\eta, \eta' \to 0$ [7]:

$$\psi(\lambda_i - i\eta, \mu_a + i\eta') = \int \frac{(\lambda_i - \lambda + i\eta)\rho_x(\lambda)}{(\lambda_i - \lambda)^2 + \eta^2} \frac{(\mu_a - \mu - i\eta')\rho_y(\mu)}{(\mu_a - \mu)^2 + \eta'^2} M(\lambda, \mu) d\lambda \, d\mu$$
$$= \int \frac{\eta\rho_x(\lambda)}{(\lambda_i - \lambda)^2 + \eta^2} \frac{\eta'\rho_y(\mu)}{(\mu_a - \mu)^2 + \eta'^2} M(\lambda, \mu) d\lambda \, d\mu + (\dots)$$
$$\underset{\eta, \eta' \to 0}{=} \pi^2 \rho_x(\lambda_i)\rho_y(\mu_a) M(\lambda_i, \mu_a) + (\dots) \tag{S31}$$

To simplify, we will assume that the population eigenvectors form a complete set of basis:

$$\mathbf{I} = \sum_{i=1}^{P} |\tilde{u}_i\rangle\langle\tilde{u}_i| = \sum_{a=1}^{P} |\tilde{w}_a\rangle\langle\tilde{w}_a|. \tag{S32}$$

Then each resolvent in Eq. (S29) can be expressed in these bases:

$$\mathbf{G}_x(z) = \sum_{i,j} |\tilde{u}_i\rangle\langle\tilde{u}_j| \, \Phi_{ij}^x(z), \quad \Phi_{ij}^x(z) : \langle\tilde{u}_i|\mathbf{G}_x(z)|\tilde{u}_j\rangle,$$
$$\mathbf{G}_y(z) = \sum_{a,b} |\tilde{w}_a\rangle\langle\tilde{w}_b| \, \Phi_{ab}^y(z'), \quad \Phi_{ab}^y(z') := \langle\tilde{w}_a|\mathbf{G}_y(z)|\tilde{w}_b\rangle, \tag{S33}$$

where $\Phi_{ij}^x$ and $\Phi_{ab}^y$ are the matrix elements of resolvents $\mathbf{G}_x(z)$ and $\mathbf{G}_y(z')$ in their respective deterministic bases. Then, Eq. (S29) simplifies to

$$\psi_P(z, z') = \mathbb{E}\left[\frac{1}{P} \sum_{i,j,a,b} \Phi_{ij}^x(z)\tilde{C}_{ja}\Phi_{ab}^y(z')\tilde{C}_{bi}^\top\right] = \frac{1}{P} \sum_{i,j,a,b} \mathbb{E}[\Phi_{ij}^x(z)]\tilde{C}_{ja}\mathbb{E}[\Phi_{ab}^y(z')]\tilde{C}_{bi}^\top, \tag{S34}$$

where we defined the deterministic overlap matrix elements $\tilde{C}_{ia} := \langle \tilde{u}_i | \tilde{w}_a \rangle$. In the second equality, we assumed that the two resolvents are independent, reducing the problem to computing the expected resolvent of a single Gram matrix.

As discussed around Eq. (S19), the resolvent $\mathbf{G}_x$ has a limiting value for $P \to \infty$ that is diagonal in the corresponding deterministic basis [8], and its matrix elements are given by:

$$\Phi^x_{ij}(z) = \frac{\delta_{ij}}{z - \tilde{\lambda}_i(1 - q_x + q_x z \mathfrak{g}_x(z))} + \mathcal{O}(P^{-1/2}), \tag{S35}$$

where $\mathfrak{g}_x(z)$ satisfies the self-consistency condition in Eq. (S11).

In order to compute the overlap $M(\lambda_i, \mu_a)$, we use Eq. (S31) and collect the term proportional to $\eta \eta'$. Thanks to Eq. (S34) and Eq. (S35), this term simplifies to:

$$\pi^2 \rho_x(\lambda_i) \rho_y(\mu_a) M(\lambda_i, \mu_a) = \frac{1}{P} \sum_{j,b} \left( \lim_{\eta \to 0} \mathrm{Im}\, \Phi^x_{jj}(\lambda_i - i\eta) \right) \tilde{C}^2_{jb} \left( \lim_{\eta' \to 0} \mathrm{Im}\, \Phi^y_{bb}(\mu_a - i\eta') \right). \tag{S36}$$

Defining

$$Q_x(\lambda_i, \tilde{\lambda}_j) := \frac{1}{\pi \rho_x(\lambda_i)} \lim_{\eta \to 0} \mathrm{Im}\, \Phi^x_{jj}(\lambda_i - i\eta), \quad Q_y(\mu_a, \tilde{\mu}_b) := \frac{1}{\pi \rho_y(\eta_a)} \lim_{\eta' \to 0} \mathrm{Im}\, \Phi^y_{bb}(\mu_a - i\eta') \tag{S37}$$

we get an equation for $M$ as

$$M(\lambda_i, \mu_a) = \frac{1}{P} \sum_{j,b} Q_x(\lambda_i, \tilde{\lambda}_j) \tilde{C}^2_{jb} Q_y(\mu_a, \tilde{\mu}_b). \tag{S38}$$

Here, $Q_x$ and $Q_y$ were already calculated in Eq. (S22). Identifying the following quantities

$$Q^x_{ij} \equiv \mathbb{E} \langle u_i | \tilde{u}_j \rangle^2 = \frac{1}{P} Q_x(\lambda_i, \tilde{\lambda}_j), \quad Q^y_{ab} \equiv \mathbb{E} \langle w_a | \tilde{w}_b \rangle^2 = \frac{1}{P} Q_y(\mu_a, \tilde{\mu}_b),$$

$$M_{ia} \equiv \mathbb{E} \langle u_i | w_a \rangle^2 = \frac{1}{P} M(\lambda_i, \mu_a), \quad \tilde{M}_{ia} := \langle \tilde{u}_i | \tilde{w}_a \rangle^2 = \tilde{C}^2_{ia}, \tag{S39}$$

we get our main result [7]:

$$\mathbf{M} = \mathbf{Q}^x \tilde{\mathbf{M}} \mathbf{Q}^{y\top},$$

$$Q^x_{ij} = \frac{1}{P} \frac{q_x \lambda_i \tilde{\lambda}_j}{\left[ \tilde{\lambda}_j(1 - q_x) - \lambda_i + q_x \lambda_i \tilde{\lambda}_j \mathfrak{h}_x(\lambda_i) \right]^2 + \left[ q_x \lambda_i \tilde{\lambda}_j \pi \rho_x(\lambda_i) \right]^2},$$

$$Q^y_{ab} = \frac{1}{P} \frac{q_y \mu_a \tilde{\mu}_b}{\left[ \tilde{\mu}_b(1 - q_y) - \mu_a + q_y \mu_a \tilde{\mu}_b \mathfrak{h}_y(\mu_a) \right]^2 + \left[ q_y \mu_a \tilde{\mu}_b \pi \rho_y(\mu_a) \right]^2}. \tag{S40}$$

### A.5 Statistics of sample eigenvalues and its concentration properties

As we discussed in the main text, the practical usage of Eq. (S22) requires computing the expectation value of individual sample eigenvalues. Eq. (S40), treating $i^{\text{th}}$ biggest sample eigenvalue as deterministic and plugging $\eta = 1/\sqrt{P}$

For sufficient conditions, we can show that the sample resolvent $\mathfrak{g}(z)$ self-averages. In this case, the sample eigenvalue density $\rho(\lambda)$ converges in law. Here, we show that for practical use of Eq. (S21), Eq. (S40), we can treat $i^{\text{th}}$ largest eigenvalue effectively as deterministic in its most probable position.

Specifically, we demonstrate that for a large number of eigenvalues $P$, the most probable $i$-th largest eigenvalue $\lambda_i$ satisfies

$$\int_{\lambda_i}^{\infty} \rho(\lambda) \, d\lambda = \frac{i}{P}, \tag{S41}$$

and that the fluctuations around this most probable is $O(1/\sqrt{P})$.

Consider a set of $P$ eigenvalues $\{\lambda_1, \lambda_2, \ldots, \lambda_P\}$ drawn independently from the probability density $\rho(\lambda)$. We order these eigenvalues in descending order:

$$\lambda_{(1)} \geq \lambda_{(2)} \geq \ldots \geq \lambda_{(P)},$$

where $\lambda_{(i)}$ denotes the $i^{\text{th}}$ largest eigenvalue. To find the most probable value $\lambda_i$ for the $i^{\text{th}}$ largest eigenvalue, we focus on the probability that *exactly* #$i$ eigenvalues exceed a threshold $\bar{\lambda}$. If we define

$$F(\bar{\lambda}) \ = \ \int_{\bar{\lambda}}^{\infty} \rho(\lambda') \, d\lambda',$$

then the probability that exactly $i$ out of $P$ samples exceed $\bar{\lambda}$ is given by the binomial expression

$$F(\bar{\lambda}, P, i) \ = \ \binom{P}{i} \left[F(\bar{\lambda})\right]^{i} \left[1 - F(\bar{\lambda})\right]^{P-i}.$$

We determine the threshold $\bar{\lambda}_i$ that maximizes $F(\bar{\lambda}, P, i)$ by setting its derivative (with respect to $\bar{\lambda}$) to zero. From this calculation, one obtains the simple condition

$$F(\bar{\lambda}_i) \ = \ \frac{i}{P}.$$

Equivalently, since $F(\lambda) = \int_{\lambda}^{\infty} \rho(\lambda') \, d\lambda'$, the most probable $i^{\text{th}}$ largest eigenvalue $\lambda_i$ satisfies

$$\int_{\lambda_i}^{\infty} \rho(\lambda) \, d\lambda \ = \ \frac{i}{P}.$$

Now we calculate approximations for fluctuation around this most probable position. Let's analyze $F(\bar{\lambda}, P, i)$ near $\bar{\lambda}_i$. Write $\bar{\lambda} = \lambda_i + \delta\lambda$ and expand $F(\bar{\lambda})$ in a Taylor series about $\lambda_i$:

$$F(\bar{\lambda}) \ = \ F(\lambda_i + \delta\lambda) \ \approx \ F(\lambda_i) \ + \ \left.\frac{dF}{d\lambda}\right|_{\lambda_i} \delta\lambda \ + \ \frac{1}{2} \left.\frac{d^2F}{d\lambda^2}\right|_{\lambda_i} (\delta\lambda)^2 \ + \ \ldots$$

Since $F(\lambda_i) = \frac{i}{P}$ and $\lambda_i$ is determined by maximizing $F(\bar{\lambda}, P, i)$, the first derivative of $F$ at $\lambda_i$ vanishes:

$$\left.\frac{dF}{d\lambda}\right|_{\lambda_i} = 0,$$

thus

$$F(\bar{\lambda}) \ \approx \ \frac{i}{P} \ + \ \frac{1}{2} F''(\lambda_i) \, (\delta\lambda)^2.$$

(We expect $F''(\lambda_i) < 0$ since $F(\lambda)$ decreases with $\lambda$.)

Substituting this expansion back into $\binom{P}{i} \left[F(\bar{\lambda})\right]^{i} \left[1 - F(\bar{\lambda})\right]^{P-i}$, we find that the dominant dependence on $\delta\lambda$ appears in a Gaussian-like factor

$$\exp\left(-\tfrac{1}{2} |F''(\lambda_i)| \, P \, (\delta\lambda)^2\right).$$

This indicates that $\bar{\lambda}$ is peaked sharply around $\lambda_i$ with a variance

$$\sigma_i^2 \ = \ \frac{1}{-F''(\lambda_i) \, P}.$$

In summary, most probable $i$-th largest eigenvalue $\lambda_{(i)}$ is determined by

$$\int_{\lambda_i}^{\infty} \rho(\lambda) \, d\lambda \ = \ \frac{i}{P},$$

with fluctuation $O(1/\sqrt{P})$.

### A.6 Statistics of sample eigenvalues and its concentration properties

Note that unlike eigenvalue density converges in law, eigenvector statistics Eq. (S21) is noisy even when $P \to \infty$ [43]. In this case, we define the $Q$ matrix as the expectation over different trials as in Eq. (S39). Equivalently, this could be obtained by averaging over a small eigenvalue interval, which could be done by plugging in a small $\eta = 1/\sqrt{P}$ to extract the pole. Note that this $1/\sqrt{P}$ is also obtained by analyzing fluctuation around the most probable $i$-th biggest eigenvalue as above. This is essentially averaging over a Cauchy distribution centered at $\lambda$ with width $\eta$. Thus, for practical usage of Eq. (S40), we simply plug this most likely $i$-th eigenvalue [8], with $\eta = 1/\sqrt{P}$.

## B Relation to regression-based similarity measures

Regression Score is not a representational similarity measure but is commonly used for scoring model closeness to the brain [46, 12]. Here, we discuss how our theoretical analysis for the overlap matrix $\mathbf{M}$ can also be applied to the regression setting. Regression score measures how well a model's activations $\mathbf{X}$ predict neural responses $\mathbf{Y}$ via a linear probe. Concretely, one performs ridge regression on a training subset $(\mathbf{X}_{1:p}, \mathbf{Y}_{1:p})$ of size $p < P$, obtaining:

$$\hat{\mathbf{X}}(p) = \mathbf{Y}\,\hat{\beta}(p). \tag{S42}$$

$$\hat{\beta}(p) = \arg\min_{\beta} \big\|\mathbf{Y}_{1:p}\beta - \mathbf{X}_{1:p}\big\|_F^2 + \alpha_{\text{reg}}\|\beta\|_F^2, \tag{S43}$$

Then the regression score gives the neural prediction error,

$$E_g(p) = \frac{\|\hat{\mathbf{X}}(p) - \mathbf{X}\|_F^2}{\|\mathbf{X}\|_F^2}, \tag{S44}$$

Note that this error can be decomposed to each error mode, where $E_g(p) = \sum_i \widetilde{W}_i(p)$ where $\widetilde{W}_i(p) := \frac{\kappa^2}{1-\gamma}\frac{W_i}{(p\lambda_i+\kappa)^2}$.

The quantity $W_i$ denotes the projection of target labels on the $i^{\text{th}}$-model eigenvalue and hence can be expressed in terms of eigencomponents, $W_i = \sum_j \frac{\lambda_j}{\sum_k \lambda_k}M_{ij}$. However, calculating $W_i$ assumes that there is access to population-level eigenvalues and poses a problem with limited data. In future work, we would like to test whether our analyses help improve the reliability of regression-based similarity methods.

## C Theory of Power-Law Spectrum

Here, we consider the case where the population spectrum obeys a power-law:

$$\tilde{\lambda}_k = \left(\frac{k}{P}\right)^{-s}, \quad k = 1, \dots, P, \quad s > 1 \tag{S45}$$

where we normalized eigenvalue indices explicitly by $P$. For large $P$, the population density becomes:

$$\tilde{\rho}(\tilde{\lambda}) = \frac{1}{P}\sum_{k=1}^{P}\delta(\tilde{\lambda} - \tilde{\lambda}_k) \sim \frac{1}{P}\int_1^P \delta(\tilde{\lambda} - \tilde{\lambda}_k)dk, \tag{S46}$$

We change the variables to $\mu := \tilde{\lambda}_k$ for which we get:

$$d\mu = -sP^s k^{-s-1}dk = -\frac{s}{P}\mu^{1+1/s}\,dk. \tag{S47}$$

In the limit $P \to \infty$, the density becomes

$$\tilde{\rho}(\tilde{\lambda}) = \frac{1}{s}\int_1^\infty \mu^{-1-1/s}\,\delta(\tilde{\lambda} - \mu)\,d\mu = \gamma\,\tilde{\lambda}^{-1-\gamma}, \quad \tilde{\lambda} \in [1,\infty], \quad \gamma = s^{-1}, \tag{S48}$$

where we defined $\gamma \in [0,1]$ for notational convenience. Note that, in this definition, the expectation value of $\tilde{\lambda}$ diverges.

## C.1 Solving the Stieltjes transform

Next, we need to solve the self-consistent equation for the Stieltjes transform Eq. (S11) which reads:

$$\mathfrak{g}(z) = \int_1^\infty \frac{\tilde{\rho}(\tilde{\lambda})}{z - \tilde{\lambda}(1 - q + qz\mathfrak{g}(z))} d\tilde{\lambda}, \quad \tilde{\rho}(\tilde{\lambda}) = \gamma\tilde{\lambda}^{-1-\gamma}. \tag{S49}$$

This integral has a closed-form solution expressed in terms of hypergeometric functions [3]. To evaluate this integral, it is convenient to work in terms of the following quantities:

$$w := \frac{z}{1-q}, \quad \beta := \frac{q}{1-q}, \quad \mathfrak{g}' := z\mathfrak{g}(z), \quad \kappa := \frac{w}{1+\beta\mathfrak{g}'}. \tag{S50}$$

Then the integral equation becomes

$$\mathfrak{g}' = \gamma\kappa \int_1^\infty \frac{\tilde{\lambda}^{-1-\gamma}}{\kappa - \tilde{\lambda}} d\tilde{\lambda} = -\gamma\kappa^{-\gamma} \int_0^\kappa \frac{u^\gamma}{1-u} du$$
$$= -\gamma\kappa^{-\gamma}\mathsf{B}(\kappa; 1+\gamma, 0), \tag{S51}$$

where we made a change of variables $u := \kappa/\tilde{\lambda}$ in the first line and used the integral definition of the incomplete Beta function

$$\mathsf{B}(z; a, b) = \int_0^z u^{a-1}(1-u)^{b-1} du. \tag{S52}$$

The integral solution reported in [3] can be obtained from Eq. (S51) by using the following identity [17] in terms of hypergeometric functions:

$$\mathsf{B}(z; a, b) = \frac{z^a(1-z)^b}{a} {}_2F_1(1, a+b; a+1; z). \tag{S53}$$

We rewrite the self-consistent equation by replacing the definition of $\kappa$

$$\mathfrak{g}' = -\gamma\left(\frac{w}{1+\beta\mathfrak{g}'}\right)^{-\gamma} \mathsf{B}\left(\frac{w}{1+\beta\mathfrak{g}'}; 1+\gamma, 0\right). \tag{S54}$$

While this equation is exact, it is not possible to solve for $\mathfrak{g}$. To obtain an analytical solution for the self-consistent equation, we need to expand the r.h.s. to leading order in $\mathfrak{g}'$:

$$\mathfrak{g}' = -\gamma w^{-\gamma}\mathsf{B}(w; 1+\gamma, 0) + \mathfrak{g}'\beta\gamma\left(\frac{w}{1-w} - \gamma w^{-\gamma}\mathsf{B}(w; 1+\gamma, 0)\right) + \mathcal{O}\left((\beta\mathfrak{g}')^2\right), \tag{S55}$$

which can be truncated to the linear order provided that $\beta\mathfrak{g}' \ll 1$. Solving for $\mathfrak{g}'$, we get

$$\mathfrak{g}' = \frac{-\gamma w^{-\gamma}\mathsf{B}(w; 1+\gamma, 0)}{1 - \beta\gamma\left(\frac{w}{1-w} - \gamma w^{-\gamma}\mathsf{B}(w; 1+\gamma, 0)\right)}. \tag{S56}$$

We also provide a power series expansion of the incomplete beta function:

$$\mathsf{B}(w; 1+\gamma, 0) = \begin{cases} \sum_{n=0}^\infty \frac{1}{n-\gamma}w^{\gamma-n} + \pi(\cot(\pi\gamma) - i), & \text{when } w \gg 1 \\ \sum_{n=1}^\infty \frac{1}{n+\gamma}w^{\gamma+n}, & \text{when } w \ll 1, \end{cases} \tag{S57}$$

which will be helpful when we implement these functions numerically. In terms of the power series, the solution becomes:

$$\mathfrak{g}' = -\gamma\frac{\pi(\cot(\pi\gamma) - i)w^{-\gamma} + \sum_{n=0}^\infty \frac{1}{n-\gamma}w^{-n}}{1 + \beta\gamma\left(\pi\gamma(\cot(\pi\gamma) - i)w^{-\gamma} + \sum_{n=0}^\infty \frac{\gamma}{n-\gamma}w^{-n} - \frac{w}{1-w}\right)}$$
$$= -\gamma\frac{\pi(\cot(\pi\gamma) - i)w^{-\gamma} + \sum_{n=0}^\infty \frac{1}{n-\gamma}w^{-n}}{1 + \beta\gamma\left(\pi\gamma(\cot(\pi\gamma) - i)w^{-\gamma} + \sum_{n=0}^\infty \frac{n}{n-\gamma}w^{-n}\right)}, \tag{S58}$$

where we simplified the denominator in the last line using $\frac{w}{1-w} = -\sum_{n=0}^\infty w^{-n}$.

Next, we compute the sample eigenvalue density $\rho(\lambda)$ and its Hilbert transform $\mathfrak{h}(\lambda)$ by computing

$$\lim_{\eta\to 0^+} \mathfrak{g}(\lambda - i\eta) = \lim_{\eta\to 0^+} \frac{\mathfrak{g}'(\lambda - i\eta)}{\lambda - i\eta} = \mathfrak{h}(\lambda) + i\pi\rho(\lambda). \tag{S59}$$

This is an extremely tedious calculation that we perform using Mathematica. Furthermore, we expand the results in $q$ and, assuming $q \ll 1$, keep only the linear term. In this regime, the leading order behavior of $\rho(\lambda)$ and $\mathfrak{h}(\lambda)$ looks like:

$$\rho(\lambda) = \gamma\lambda^{-1-\gamma}\left(1 - q\gamma\left(2\pi\gamma\cot(\pi\gamma)\lambda^{-\gamma} + \sum_{n=1}^{\infty}\frac{n+\gamma}{n-\gamma}\lambda^{-n}\right)\right) + \mathcal{O}(q^2)$$

$$\mathfrak{h}(\lambda) = \lambda^{-1}\left(1 - \lambda^{-\gamma}\pi\gamma\cot(\pi\gamma) - \lambda^{-1}\frac{\gamma}{1-\gamma}\right)$$

$$+ \pi\gamma^2 q\left(\pi\gamma\lambda^{-2\gamma-1}\left(\cot^2(\pi\gamma) - 1\right) + \lambda^{-\gamma-2}\frac{(\gamma+1)\cot(\pi\gamma)}{1-\gamma}\right) + \mathcal{O}(q^2, \lambda^3). \tag{S60}$$

Here, we did not include higher-order terms for $\mathfrak{h}(\lambda)$ to avoid clutter.

Finally, we use the formula for estimating sample eigenvalues Eq. (S41) for which we obtain an explicit formula:

$$\mathfrak{F}(\lambda, q; \gamma) := \int_\lambda^\infty \rho(\lambda)d\lambda = \lambda^{-\gamma}\left(1 - q\gamma^2\left(\lambda^{-\gamma}\pi\cot\pi\gamma + \sum_{n=1}^{\infty}\frac{1}{n-\gamma}\lambda^{-n}\right)\right). \tag{S61}$$

Here, the semi-colon separates sample-related arguments that we have empirical access to $(\lambda_i, q)$ from the only population-related quantity, $\gamma$. Hence, using the following relation [34, 8]

$$\mathfrak{F}(\lambda_i, q; \gamma) = \frac{i}{P} \tag{S62}$$

we can either predict the shape of empirical eigenvalues given the decay rate of population spectrum (forward), or infer the population decay rate given the empirical observations of eigenvalues (backward). Finally, we numerically test our theory and obtain perfect agreement with empirical data in Fig. S1.

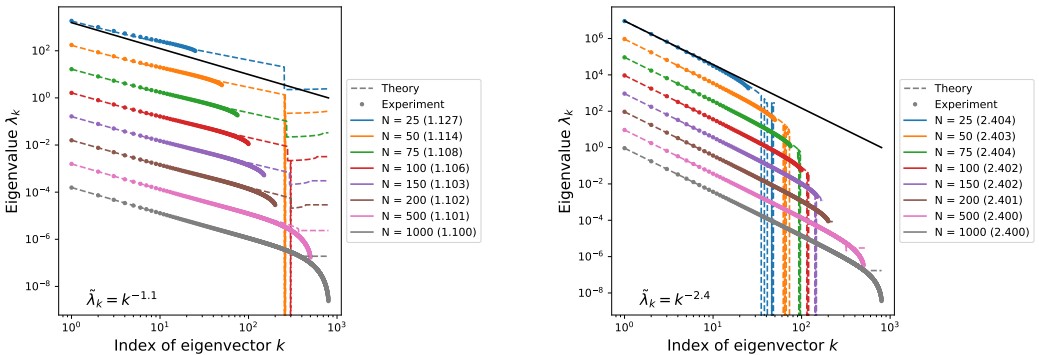

Figure S1: For a population spectrum with $\tilde{\lambda}_k = k^{-1.1}$ (Left) and $\tilde{\lambda}_k = k^{-2.4}$ (Right), we show the spectra of the empirical eigenvalues for different $N$. Black solid line indicates the true eigenvalue decay. The numbers in parentheses in the legend indicate the inferred true decay rate from a population of $N$. In the regime $s < 2$ ($\gamma > 0.5$), the empirical eigenvalues are always overestimated (Left), and in the regime $s \geq 2$ ($\gamma < 0.5$) they are always underestimated (Right).

## D  Experimental Details

Code for all experiments is publicly available in this Github repository. All experiments were done using a single A100 GPU.

### D.1 Synthetic Data

For the synthetic experiments, we generate a population activation matrix in $\mathbb{R}^{P \times \tilde{N}}$ whose Gram matrix follows a chosen spectral distribution (e.g., a power-law). We then form the sample activation matrix by projecting onto a random subset (or random linear subspace) of size $N$, yielding $\mathbb{R}^{P \times N}$. This procedure enables us to directly control the underlying population eigenvalues and eigenvectors, facilitating clean comparisons between sample-level and population-level similarity measures.

### D.2 Brain Data

We employ a set of publicly available neural recordings from primate visual cortex (e.g., V2) and compare these against the representations of various vision models, similarly to the methodology in [12]. In total, we evaluate 32 models spanning supervised, self-supervised, and adversarially trained architectures, including well-known families such as ResNet, DenseNet, MobileNet, EfficientNet, and Vision Transformers. We extract intermediate-layer activations for each model on the same set of visual stimuli used in the neural recordings, applying the standard preprocessing routines (e.g., image resizing, ImageNet normalization).

Within each model, we select one or more representative layers (e.g., post-ReLU or transformer blocks). We then compute Gram matrices from those activations, matching the dimensionality of the neural dataset. In scenarios where the dataset contains more neurons than we wish to analyze, we project the data into a lower-dimensional subspace of size $N$. Finally, we compute representational similarity (e.g., CKA or (SV)CCA) between these model-derived Gram matrices and the neural Gram matrices, both in their raw (sample) forms and using our denoising procedure for backward inference.

## E   Another denoising method: truncated inverse

We utilize a truncated Singular Value Decomposition (SVD) to obtain a regularized estimate of $\tilde{\mathbf{M}}$:

$$\tilde{\mathbf{M}} = \mathbf{V}\boldsymbol{\Sigma}_{\text{trunc}}^{-1}\mathbf{U}^\top\mathbf{M}, \tag{S63}$$

where $\mathbf{Q}^{(x)} = \mathbf{U}\boldsymbol{\Sigma}\mathbf{V}^\top$ is the SVD of $\mathbf{Q}^{(x)}$, and $\boldsymbol{\Sigma}_{\text{trunc}}^{-1}$ is the truncated inverse of the singular values, defined as:

$$\left(\boldsymbol{\Sigma}_{\text{trunc}}^{-1}\right)_{ii} = \begin{cases} \frac{1}{\sigma_i} & \text{if } i \leq \tau, \\ 0 & \text{otherwise,} \end{cases} \tag{S64}$$

## F   Sample CKA with population eigenvalue term

We demonstrate that eigenvector delocalization is the dominant factor causing the decrease in sample CKA for rapidly decaying spectra such as power-law or exponential distributions.

In Fig. S2, we consider a population with eigenvalues following $\tilde{\lambda}_i = i^{-1.2}$ and in Fig. S3, we examine a population with eigenvalues following $\tilde{\lambda}_i = e^{-i}$. These experiments confirm that while sampling affects both eigenvalues and eigenvectors, the systematic underestimation of CKA is predominantly caused by eigenvector delocalization rather than changes to the eigenvalue distribution. This holds true for both power-law and exponential eigenvalue spectra, which are common in neural data.

## G   Confidence Interval under Maximum Likelihood Estimation

Although it is hard to calculate the uncertainty of the estimator from constrained optimization, we can compute it using maximum likelihood estimation for $\tilde{M}_{ja}$.

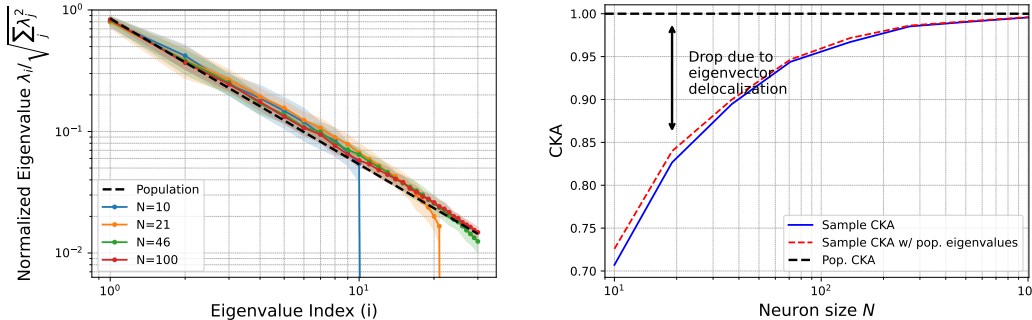

Figure S2: **Left:** Normalized sample eigenvalues for $P = 100$ and $\tilde{\lambda}_i = i^{-1.2}$ with varying $N$. The first few terms of normalized eigenvalues remain relatively stable despite neuron sampling. **Right:** Sample CKA between brain and model with identical representations (true CKA = 1) as neurons are sampled. The deviation is primarily due to eigenvector delocalization, as shown by the close match between observed sample CKA and CKA calculated with population eigenvalues but sample eigenvectors.

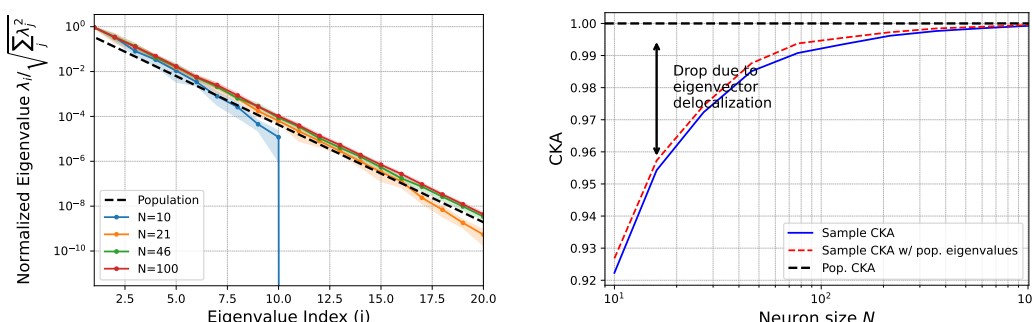

Figure S3: **Left:** Normalized sample eigenvalues for $P = 100$ and $\tilde{\lambda}_i = e^{-i}$ with varying $N$. As with the power-law case, the dominant normalized eigenvalue components remain relatively stable under sampling. **Right:** Sample CKA behavior with exponential eigenvalue decay. The observed sample CKA closely tracks the hybrid CKA (population eigenvalues with sample eigenvectors), confirming that eigenvector delocalization primarily drives the CKA reduction.

### G.1  Problem Setup

For two sets of eigenvectors $\{|u_j\rangle\}_{j=1}^{P}$ (unobserved) and $\{|v_a\rangle\}_{a=1}^{P}$ (observed), we estimate confidence intervals for $\tilde{M}_{ja} = \left| \langle u_j | v_a \rangle \right|^2$ based on empirical eigenvectors $\{|\hat{u}_i^{(t)}\rangle\}_{i=1}^{P}$ across trials $t$.

Our neuron-wise sampling model assumes:

$$|\hat{u}_i^{(t)}\rangle = \sum_{j=1}^{P} \epsilon_{ij}^{(t)} \sqrt{Q_{ij}} |u_j\rangle , \quad \epsilon_{ij}^{(t)} \sim \mathcal{N}(0,1) \text{ i.i.d.} \tag{S65}$$

where $Q \in \mathbb{R}^{P \times P}$ is fixed across trials.

Projecting onto $|v_a\rangle$ gives:

$$\left\langle v_a \middle| \hat{u}_i^{(t)} \right\rangle \sim \mathcal{N}(0, \sigma_{ia}^2), \quad \sigma_{ia}^2 = \sum_{j=1}^{P} Q_{ij} \tilde{M}_{ja} \tag{S66}$$

The squared overlaps follow:

$$M_{ia}^{(t)} \equiv \left| \left\langle v_a \middle| \hat{u}_i^{(t)} \right\rangle \right|^2 \sim \sigma_{ia}^2 \chi_1^2 \tag{S67}$$

## G.2 Confidence Intervals (CI)

### G.2.1 Single-Trial Case

The negative log-likelihood for a single trial is:

$$\ell_a(\tilde{M}_{\cdot a}) = \frac{1}{2} \sum_{i=1}^{P} \left[ \ln \sigma_{ia}^2 + \frac{M_{ia}}{\sigma_{ia}^2} \right] \tag{S68}$$

For a confidence interval on $\tilde{M}_{ka}$, we use profile likelihood:

$$L_{\text{profile}}(m) = \max_{\{\tilde{M}_{ja}: j \neq k, \, 0 \leq \tilde{M}_{ja} \leq 1\}} \exp\left(-\ell_a(\tilde{M}_{\cdot a})\right) \quad \text{with } \tilde{M}_{ka} = m \tag{S69}$$

The $(1 - \alpha)$ CI is:

$$\text{CI}_{1-\alpha}(\tilde{M}_{ka}) = \{m \in [0,1] : -2\log\left(L_{\text{profile}}(m)/L_{\max}\right) \leq \chi_{1,1-\alpha}^2\} \tag{S70}$$

---

**Algorithm 2** Profile CI for $\tilde{M}_{ka}$ (Single Trial)

---

1: **Input:** $M_{ia}$, $Q$, index $k$, level $1 - \alpha$
2: **Output:** $[\tilde{M}_{ka}^{\text{lower}}, \tilde{M}_{ka}^{\text{upper}}]$
3: Set $\tau = \chi_{1,1-\alpha}^2$. Compute $L_{\max}$ by minimizing $\ell_a$ in Eq. (S68).
4: **for** $m$ on a grid in $[0,1]$ **do**
5:     Minimize $\ell_a(\tilde{M}_{\cdot a})$ subject to $\tilde{M}_{ka} = m$ and $0 \leq \tilde{M}_{ja} \leq 1$ ($j \neq k$).
6:     Set $\Lambda(m) = -2\log\left(L_{\text{profile}}(m)/L_{\max}\right)$.
7: **end for**
8: Return the smallest and largest $m$ with $\Lambda(m) \leq \tau$.

---

### G.2.2 Multiple Trials

For $T$ trials with the same $Q$, the joint negative log-likelihood is:

$$\ell_a^{\text{joint}}(\tilde{M}_{\cdot a}) = \sum_{t=1}^{T} \frac{1}{2} \sum_{i=1}^{P} \left[ \ln \sigma_{ia}^2 + \frac{M_{ia}^{(t)}}{\sigma_{ia}^2} \right] \tag{S71}$$

The joint profile likelihood and CI follow analogously:

$$\text{CI}_{1-\alpha}^{(T)}(\tilde{M}_{ka}) = \{m \in [0,1] : \Lambda_T(m) \leq \chi_{1,1-\alpha}^2\} \tag{S72}$$

---

**Algorithm 3** Profile CI for $\tilde{M}_{ka}$ (Multiple Trials)

---

1: **Input:** $\{M_{ia}^{(t)}\}_{t=1}^{T}$, $Q$, index $k$, level $1 - \alpha$, optional weights $w_t$
2: **Output:** $[\tilde{M}_{ka}^{\text{lower}}, \tilde{M}_{ka}^{\text{upper}}]$
3: Set $\tau = \chi_{1,1-\alpha}^2$. Compute $L_{\max}^{(T)}$ by minimizing $\ell_a^{\text{joint}}$.
4: **for** $m$ on a grid in $[0,1]$ **do**
5:     Minimize $\ell_a^{\text{joint}}(\tilde{M}_{\cdot a})$ subject to $\tilde{M}_{ka} = m$ and $0 \leq \tilde{M}_{ja} \leq 1$ ($j \neq k$).
6:     Set $\Lambda_T(m) = -2\log\left(L_{\text{profile}}^{(T)}(m)/L_{\max}^{(T)}\right)$.
7: **end for**
8: Return the smallest and largest $m$ with $\Lambda_T(m) \leq \tau$.

---

## G.3 CIs for CKA and CCA

For weighted functionals like CKA or CCA:

$$f(\tilde{M}) = \sum_{j=1}^{P} \sum_{a=1}^{P} w_j w_a' \tilde{M}_{ja} \tag{S73}$$

Apply the delta method:

1. Compute gradient: $\frac{\partial f}{\partial \tilde{M}_{ja}} = w_j w_a'$

2. Estimate Fisher information: $\mathcal{I}_{\hat{\tilde{M}}} = \nabla^2 \ell^{\text{joint}}(\hat{\tilde{M}})$

3. Calculate variance: $\mathrm{Var}(f) = \nabla f^\top \mathcal{I}_{\hat{\tilde{M}}}^{-1} \nabla f$

4. CI: $f(\hat{\tilde{M}}) \pm z_{1-\alpha/2} \cdot \sqrt{\mathrm{Var}(f)}$

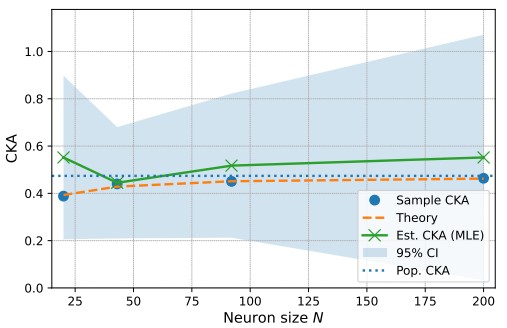 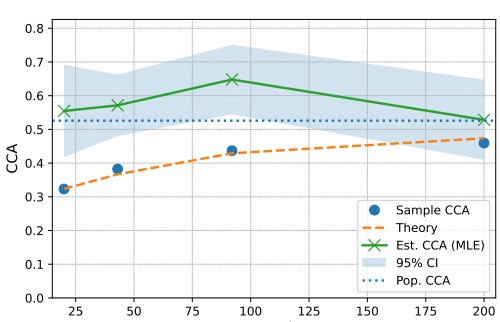

Figure S4: MLE estimation and confidence interval of 5 trials. Here $P = 100$, and the two population Gram matrices had power-law eigenvalues with exponent $-1.2$. **Left:** For CKA. Blue dots are the empirical sample CKA, where the orange dotted line is the theoretical line for sample CKA. The green solid line is the estimated population CKA using MLE, and the blue shades are the $95\%$ confidence interval. **Right:** Same analysis but for CCA.

## H Eigenvector Delocalization via Support Mergers

### H.1 Setup and diagnostic

Let $\Sigma \in \mathbf{R}^{P \times P}$ have eigenvalues $\{\tilde{\lambda}_i\}_{i=1}^{P}$. From $N$ i.i.d. samples $X = \Sigma^{1/2} Z$ with $Z_{ij} \sim \mathcal{N}(0, 1)$, the sample covariance is $S = \frac{1}{N} X X^\top$. A convenient diagnostic for support components is

$$\mathcal{B}(x) := \frac{1}{x} + \frac{1}{N} \sum_{i=1}^{P} \frac{1}{\frac{1}{\tilde{\lambda}_i} - x}, \tag{S74}$$

whose monotonicity between its poles $x_i = 1/\tilde{\lambda}_i$ tracks gaps vs. support. Intervals with $\mathcal{B}'(x) < 0$ correspond to gaps; support edges occur where the local monotonicity changes (tangency/inflection). See [8] for related transform pictures.[5]

### H.2 From support mergers to eigenvector delocalization

Decreasing $N$ is akin to increasing an effective noise level: nearby spikes mix first (Dyson Brownian motion intuition). Thus leading (well-separated) spikes remain isolated, while deeper ones merge into a common bulk. Empirically, the diagonal overlaps $Q_{ii}$ stay near 1 up to an index $i'$ and then drop sharply; the drop aligns with the point where a *local* neighborhood can no longer sustain two separated support components.

---

[5]Sign conventions vary; our conclusions are invariant under a global sign flip.

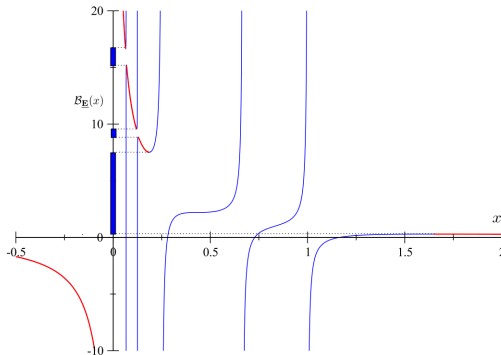

Figure S5: **Counting components via $\mathcal{B}$.** Blue segments on the vertical axis correspond to support intervals (adapted from Fig. 6 of [8]).

$$\mathrm{d}\lambda_i = \sqrt{\frac{2}{\beta N}}\mathrm{d}B_i + \frac{1}{N}\sum_{j=1}^{N}\frac{\mathrm{d}t}{\lambda_i - \lambda_j}$$

$$\mathrm{d}\mathbf{v}_i = \frac{1}{\sqrt{N}}\sum_{\substack{j=1\\j\neq i}}^{N}\frac{\mathrm{d}B_{ij}}{\lambda_i - \lambda_j}\mathbf{v}_j - \frac{1}{2N}\sum_{\substack{j=1\\j\neq i}}^{N}\frac{\mathrm{d}t}{(\lambda_i - \lambda_j)^2}\mathbf{v}_i$$

Figure S6: **Heuristic.** Additive Dyson Brownian motion shown; in our multiplicative setting, the same local-mixing picture applies: smaller inter-spike spacing $\lambda_i - \lambda_{i+1} \Rightarrow$ earlier merger.

## H.3 Two-peak approximation and the $\sqrt{N}$ law

Assume a power-law population spectrum

$$\tilde{\lambda}_i = i^{-1-\gamma}, \qquad \gamma > 0, \tag{S75}$$

so the poles of $\mathcal{B}$ are $x_i = 1/\tilde{\lambda}_i = i^{1+\gamma}$. Between two consecutive poles $b = x_i$ and $a = x_{i+1}$ ($a > b$), approximate locally

$$\mathcal{B}_{\mathrm{loc}}(x) \approx \frac{1}{x} + \frac{1}{N}\left[\frac{1}{a-x} + \frac{1}{b-x}\right]. \tag{S76}$$

A local merger occurs when $x \mapsto \frac{1}{x}$ is tangent to the rational term (equivalently, equal value and slope at $x^\star \in (b,a)$). Solving the tangency system gives the critical sample size

$$N_i^\star = \frac{\left[(i+1)^{\frac{2}{3}(1+\gamma)} + i^{\frac{2}{3}(1+\gamma)}\right]^3}{\left[(i+1)^{1+\gamma} - i^{1+\gamma}\right]^2}. \tag{S77}$$

For $N > N_i^\star$, the two local components near $x_i$ and $x_{i+1}$ remain separated; for $N < N_i^\star$, they merge.

*Remark* H.1 (**Key: $\sqrt{N}$ scaling of eigenvector delocalization**). Using $(i+1)^{1+\gamma} - i^{1+\gamma} = (1+\gamma)i^\gamma + o(i^\gamma)$ and $(i+1)^{\frac{2}{3}(1+\gamma)} + i^{\frac{2}{3}(1+\gamma)} = 2i^{\frac{2}{3}(1+\gamma)} + o(i^{\frac{2}{3}(1+\gamma)})$, Eq. (S77) yields

$$N_i^\star = \frac{8}{(1+\gamma)^2}i^2[1 + o(1)].$$

Hence the delocalization threshold (where $Q_{ii}$ drops) occurs at

$$i^\star(N) \asymp \frac{1+\gamma}{\sqrt{8}}\sqrt{N}.$$

*Interpretation:* the number of population-aligned eigenvectors grows only like $\sqrt{N}$; beyond $i^\star$, local support components have merged and the corresponding sample eigenvectors are mixed with the bulk. This $\sqrt{N}$ law is the main practical takeaway. See Fig. S7.

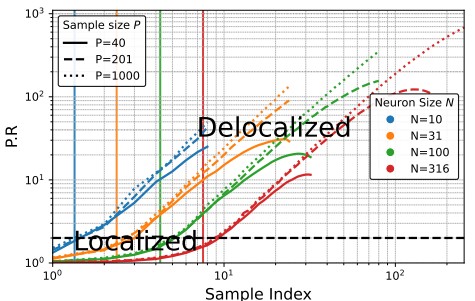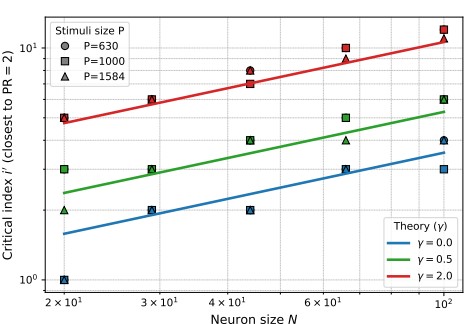

Figure S7: **Left:** PR of sample eigenvector when population eigenvalue followed power-law with exponent $-1.2$. Vertical lines with colors are theoretical predictions for the critical index from the two-peak approximation. The black horizontal dashed line corresponds to PR = 2, which roughly marks the index at which the eigenvector becomes delocalized. **Right:** Critical index $i'$, where population eigenvalue followed power-law with exponent $-1 - \gamma$. Solid lines are theoretical predictions from the two-peak approximation, $i' = \frac{1+\gamma}{\sqrt{8}}\sqrt{N}$. Markers are empirical critical index when PR is 2.

