# OpenReview forum: "Spectral Analysis of Representational Similarity with Limited Neurons"
_NeurIPS.cc/2025/Conference — NeurIPS 2025 poster_

### Official Review · Reviewer_HcPT · 2025-06-20

**Clarity:** 2
**Significance:** 3
**Originality:** 3
**Rating:** 5
**Confidence:** 4

**Summary:**

This paper studies how sampling a subset of neurons in a large population (modeled as Gaussian projection) affects estimates of the alignment between the sampled representation and some fixed reference population. Its main result applies existing characterizations of eigenvector overlaps of Wishart matrices to describe how eigenvector delocalization induces bias in plug-in estimators of representational alignment. The authors then leverage this characterization to develop a corrected estimator.

**Questions:**

1. The main technical limitation I see in this paper is that the authors do not provide a full characterization of the bias of their proposed algorithm. The justification provided in 4.2.2 for why it works is heuristic in nature. It would be useful if, even just for the setting of power law data, you could give a more complete theoretical analysis of when the algorithm works. For instance, fixing a particular population decay exponent, how many components does one have to accurately estimate in order to estimate the CKA/CCA within some desired error tolerance, and when can the constrained optimization used to estimate $\tilde{M}$ succeed in estimating that number of components? How does the choice of algorithms used to estimate the power law exponent and the overlap matrix affect the final estimate of the CKA/CCA? The lack of clarity around this issue is my main reason for recommending rejection.

2. The authors mention moment-based unbiased estimators for the CKA in lines 231-235, but do not compare these methods to their proposed estimator. Adding a few such comparisons would help the reader choose a method. In particular, it would be useful to include those alternative estimators in Figure 4.

3. The series of approximations made in Appendix C is not clearly motivated. Can you elaborate on why each of the assumptions made in order to render the computation tractable make sense in the context of the neuroscience data of interest?

4. I have several formatting suggestions, which I think would clarify the presentation:

    1. I think it would be helpful if the authors stated their main theoretical result (deterministic equivalents for CCA and CKA) as a proposition or theorem, since as written the lede is somewhat buried in Section 3 (the main result is stated in Lines 134-135 as "Plugging in the formula for expected $M_{ia}$ in Eq. 5 and Eq. 6, we get an analytical formula for CCA and CKA, respectively").

    2. The paper employs somewhat non-standard notation for sample and population covariance matrices, denoting them by $\Sigma$ and $\tilde{\Sigma}$, respectively. Unless the authors have a good reason for not doing so, I suggest that they adopt the more standard notation of $\hat{\Sigma}$ and $\Sigma$, where quantities estimated from samples are distinguished from their population counterparts by a circumflex.

**Ethical Concerns:**

["NO or VERY MINOR ethics concerns only"]

**Final Justification:**

The authors have done a good job of addressing my concerns, and have adequately responded to the other referees. I have given a longer description of why I favor acceptance in a comment below.

**Limitations:**

On the whole, I think there is room for the authors to better address the limitations of their work. One point I wish they remarked upon more clearly is the fact that they do not consider the effect of random stimulus sampling; a sentence or two in the discussion would suffice. Some more prose could also be devoted to the limitations of the proposed algorithm.

**Quality:**

2

**Strengths And Weaknesses:**

On the whole, I think that this paper is timely, and will certainly be of interest to the community of researchers working on representational alignment. In my view, its main weakness is that the connection between the theoretical results and the proposed algorithm for obtaining corrected estimates of alignment is not so strong. For instance, so far as I can tell the authors do not have a guarantee that it produces an asymptotically unbiased estimate. This is an important limitation because the authors make quite a bit of the fact that their approach comes with more theoretical insight relative to unbiased estimators based on moment computations (Line 233). Most of my questions below are focused on this point. I also have a few suggestions for how the authors could improve the clarity of their manuscript, but those are minor concerns in comparison.

---

> ### Author Rebuttal · Authors · 2025-07-31
>
> We thank the reviewer for their thorough review. We hope our response addresses the reviewer’s concerns and look forward to discussing it further.
>
> **Strength and Weaknesses:**
>
> ***(Theoretical Insight)*** Moment-based unbiased estimators for CKA are plug-in estimators that correct for finite-sample bias arising from both input sampling and neuron sampling. While these methods are useful in practical settings, they offer limited theoretical insight—for example, they do not theoretically reveal how CKA varies with the number of neurons or how the eigenspectrum structure influences the finite-sample CKA.
>
> Our work is complementary to these methods in that it enables an analytical understanding of representational similarity, specifically in the regime of limited neuron sampling. For example, consider recording $N$ neurons from the brain. An unbiased estimator can correct for finite-sample bias when comparing neural recordings to a model, ideally yielding a value close to the true similarity. In contrast, the random matrix theory allows one to analyze the spectral properties of the sample Gram matrix to infer, for instance, how many additional neurons are needed to resolve further eigenvectors. This type of inference is not possible with plug-in estimators.
>
> ***(Asymptotic unbiasedness)*** We are grateful to the reviewer for raising this issue, which led us to improve our theoretical analysis. Here, we discuss the bias in our proposed algorithm and will provide a more detailed analysis in our manuscript.
>
> The random matrix theory, which underlies our analysis, is valid in the limit $P, N \to \infty$ ($q = P/N \in (0,\infty)$) and is asymptotically unbiased by construction. In practice, for finite $N$ , the theoretical prediction for the self-overlap matrix $Q$ has order $1/\sqrt{N}$ fluctuations (see Ref A). Propagating this error to the estimated CKA, we also expect a similar $\mathcal{O}(1/\sqrt{N})$ fluctuation. We will provide additional experiments to demonstrate this.
>
> In summary, we emphasize again that our method provides a predictive theory of CKA and complements unbiased estimators. In practice, we believe unbiased estimators are useful, especially for very small data, and should always be compared to RMT predictions. As the reviewer rightfully pointed out, the current version lacks a comparison between two methods, and we now include unbiased estimators in our experiments (see below).
>
> In Section 4, we will elaborate on this in more detail and also include a separate subsection discussing the differences with unbiased estimators. We are also re-running all our experiments with unbiased estimators and will include the plots in the revised version (see below).
>
> [a] Bun et al., On the overlaps between eigenvectors of correlated random matrices
>
>
> **Questions:**
>
> **1:** *(illustrative example)*
>
> Consider two identical representations whose population eigenvalues follow a power law with exponent $\gamma = -1.2$ and $P = 100$. In this setting, the top $5 \times 5$ block of
> $
> \frac{\tilde{\lambda_i}}{\sqrt{\sum_{j=1}^P \tilde{\lambda_j}^2}} \cdot
> \frac{\tilde{\mu_a}}{\sqrt{\sum_{b=1}^P \tilde{\mu_b}^2}} \cdot
> \tilde{M_{ia}}
> $
> already accounts for about 95\% of the CKA. While this is a simple toy case, the takeaway is that when eigenvalues decay quickly, the leading components dominate the CKA; with sufficiently many recorded neurons, these are precisely the localized components that can be estimated accurately.
>
> *(From heuristic to quantitative uncertainty.)*
>
> We agree that the above argument is still heuristic. Rather than attempting a full bias analysis on constrained optimization, we provide confidence intervals for $\tilde{M_{ia}}$ using a maximum likelihood estimation (MLE) approach.
>
> First assume that we had the correct parametric form of the population eigenvalues, so that we estimated them accurately and thus obtained the self-overlap matrix $Q$ (as shown in Appendix C). Under a Gaussian ansatz(see Ref a where
> $
> \ket{u_i^{(t)}} = \sum_{j=1}^P \epsilon_{ij}^{(t)} \sqrt{Q_{ij}} \ket{\tilde{u_j}}
> \quad \text{with} \quad \epsilon_{ij}^{(t)} \sim \mathcal{N}(0,1),
> $
> (see [Ref A])
> the squared overlaps follow $M_{ia}^{(t)} \sim \sigma_{ia}^2 \chi_1^2$, where $\sigma_{ia}^2 = \sum_{j=1}^P Q_{ij} \tilde{M_{ja}}$.
>
> Using profile likelihood on the negative log-likelihood
> $
> \ell_a(\tilde{M_{\cdot a}}) = \frac{1}{2} \sum_{i=1}^P \left[ \ln \sigma_{ia}^2 + \frac{M_{ia}}{\sigma_{ia}^2} \right],
> $
> and running MLE, we obtain $(1 - \alpha)$ confidence intervals that translate directly to CKA/CCA via the delta method. (We added an Appendix for algorithms and also comparisons between true similarity, MLE estimates, and confidence intervals.)
>
> In practice, we used constrained optimization rather than unconstrained MLE since each matrix element lies in $[0, 1]$, which yields slightly more stable results in extremely small neuron regimes (e.g., $N = 10$).
>
> *(How many components are well localized.)*
>
> We quantify the number of well-localized eigenvectors by analyzing when sample eigenvalue support components merge. For power-law population spectra $\tilde{\lambda_i} = i^{-1 - \gamma}$, we use a local two-peak approximation of the blue function
> $
> \mathfrak{B}(x) = \frac{1}{x} + \frac{1}{N} \sum_{i=1}^P \frac{1}{\frac{1}{\tilde{\lambda_i}} - x}
> $
> (see [Ref b]) to derive the critical neuron size $N_i^\star$ at which adjacent components coalesce.
>
> This yields a delocalization threshold $i^\star \sim \frac{1 + \gamma}{\sqrt{8}} \sqrt{N}$, implying that only $O(\sqrt{N})$ leading eigenvectors remain reliably localized — precisely those whose diagonal overlaps $Q_{ii}$ stay near 1 before dropping sharply at $i^\star$.
>
>
> [a] Monasson et al., *Estimating the spectrum of large covariance matrices from random subsamples*
>
> [b] Bun et al., *Cleaning large correlation matrices*
>
>
>
>
> **2:**
> We thank the reviewer for this suggestion. We now include the CKA predictions using unbiased estimators in all our experiments. Our method almost perfectly agrees with the unbiased estimator in synthetic dataset experiments.
>
> In Fig.4, we found that our method tends to slightly overestimate CKA compared to the unbiased estimator, but it does not change the overall picture. We will include these figures in our appendix. We now confirm this behavior also for other regions (V1, V4, IT), and will include their plots in the appendix.
>
>
>
>
> **3.:** The approximation $\beta \mathfrak{g}' \ll 1$ is mainly used to obtain an analytical solution (see ref.[3]). It essentially corresponds to the large eigenvalue $\lambda \gg 1$ ***and*** small $q\ll 1$ limit.
>
> To see this, we note that $\mathfrak{g} \sim 1/z$ for $z\gg1$. Since  $\mathfrak{g}'=z\mathfrak{g} \sim \mathcal{O}(1)$ remains order 1, the condition $\beta \mathfrak{g}' \ll 1$ implies $\beta \sim q\ll 1$. This can be also seen from Eq.S60 where the small $q$ correction to $\rho(\lambda)$ is given by an infinite power series in terms of $\lambda^{-1}$, which is negligible for large eigenvalues.
>
> The limit $\lambda \gg 1$ can be relevant to neural data analysis since CKA is dominated by large eigenvalues, and the limit $q\ll 1$ is relevant when we study small deviations from the population density. Furthermore, we numerically confirm that our final formula leads to precise results for the power-law distribution, which seems to be relevant in many neuroscience experiments.
>
> We will include a detailed explanation of the reasoning behind our assumptions in Appendix C.
>
>
>
>
> **4.: ** (a) We thank the reviewer for this suggestion. We now state our main result as a proposition. (b) We made this choice because we mainly work with sample quantities, and with the hat, the equations looked too crowded. We can certainly change the notation if it would help readability.
>
>
>
>
> **Limitations:** We thank the reviewer for pointing this issue out. Some works handle this using estimators, but an RMT treatment seems to be difficult and beyond the scope of this work. We will explicitly mention this issue in a separate Limitations section.

---

> ### Comment · Reviewer_HcPT · 2025-08-02
>
> I thank the authors for their thorough reply to my questions. I think my major concerns have been satisfactorily addressed, and I will increase my score accordingly to a 5 when submitting my final recommendation. Given that I intend to increase my score, I would like to elaborate on why, though I think some of Reviewer iYBp's concerns are well-founded, I am still in favor of acceptance.
>
> First, regarding the assumption of Gaussian random projections: I of course agree that the assumptions under which randomly-oriented projections and subsampling are equivalent are clearly not always satisfied in neural data, and that this inequivalence is an issue deserving of more theoretical attention (going back to Gao and Ganguli's "A theory of multineuronal dimensionality, dynamics and measurement"). However, I will echo the authors in offering a random matrix theorist's standard apologia for considering rotation-invariant ensembles: given an estimation problem of interest, it is worth first bringing the powerful analytical toolkit available assuming rotation-invariance to bear to see what surprises arise even in this highly symmetric, idealized world. Then, one can subsequently dive into the world of non-invariant ensembles, through what is usually a gradual process based on exploration of specific models for non-invariant random matrices. Identifying when and how the predictions of results derived assuming rotation-invariance break down is a useful part of in this process, and requires the results of the rotation-invariant computation to be known. Thus, I am not inclined to dismiss the results of this paper out of hand because the authors have assumed Gaussian projections.
>
> Second, regarding whether static measures of representational similarity like CCA/CKA are worthy of study at all: The fact remains that a portion of the neuroscience community uses static measures to compare representations, no? (I would also strongly contest the claim that the neuroscience community at large has overlooked dynamics, but that is neither here nor there) Pragmatically, for those that do so---irrespective of whether the application of this static measure is appropriate for the question of interest---it seems worth knowing how their results are biased. Respectfully, I do not think that it is fair to place the blame for the sins (if they are indeed so) of some fraction of the broader community solely upon the shoulders of the authors of the present manuscript. Moreover, in the same way as I see the rotation-invariant assumptions of this manuscript as a first step towards an understanding of subsampling, I would see theoretical characterization of static measures as a first step towards an understanding of those measures that account for dynamics. I look forward to further discussion of this issue with the authors, and with Reviewer iYBp.

---

> > ### Author Response · Authors · 2025-08-04
> >
> > We are glad to be able to address the reviewer's questions, and we thank them for their constructive comments. We agree with both Reviewer HcPT and iYBp that linear representational similarity measures inherently overlook non-linear relationships in data.
> >
> > We are accordingly writing a separate "Limitations" section, which thoroughly discusses:
> >
> > - The assumption of Gaussian (linear) projections and their potential pitfalls when analyzing neural data.
> >
> > - The fact that all similarity measures are constructed based on certain symmetry assumptions (linear or nonlinear), and the current method only accounts for rotational symmetry and ignores other neuroscientifically relevant symmetries.
> >
> > We also thank the reviewer for engaging in this stimulating discussion of representational similarity measures. As the reviewer correctly pointed out, our main goal in this work is to analyze a widely used similarity measure and check whether its applications may potentially lead to incorrect results (also please see our response to reviewer iYBp).
> >
> > Additionally, similarity measures serve as summary statistics for various aspects of high-dimensional representational geometry. Developing such fine-grained analytical tools enables us to decompose these measures into their spectral components and identify which geometric features dominate the similarity. We hope that these studies will ultimately allow us to design more principled similarity metrics.

---

> ### Comment · Reviewer_HcPT · 2025-08-05
>
> Thanks for following up! I think the addition of a formal "limitations" section will be helpful, and I like the idea of highlighting that one should choose a similarity measure with appropriate symmetries (especially when it comes to considering representational drift and related issues). Though I gather it's not yet visible to you as authors, I have raised my score.

---

### Official Review · Reviewer_fmbc · 2025-06-22

**Clarity:** 3
**Significance:** 4
**Originality:** 3
**Rating:** 5
**Confidence:** 4

**Summary:**

This paper demonstrates that in certain theoretically tractable settings, subsampling neural populations systematically underestimates neural similarity.

**Questions:**

1. From line 89, it appears that subsampling is modeled using a random projection matrix $R$. However, it's not immediately clear to me why this is an appropriate model for subsampling. Intuitively, I would expect subsampling to involve selecting specific units rather than mixing information across all units, as a random Gaussian projection does. Could you clarify the rationale behind this modeling choice?
2. In the literature, one typically encounters similarity metrics which are normalized by the maximum explainable variance (i.e., similarity is measured relative to a noise ceiling). See, for example, equation 1 [here](https://arxiv.org/pdf/2502.16238). If you were to use this metric in your setting, would you still be systemically underestimating the neural similarity?
3. There has been much recently work on similarity metrics between dynamical systems, starting with [DSA](https://arxiv.org/pdf/2306.10168) but see also [this](https://arxiv.org/abs/2410.24070) work. Do you think your theoretical insights would carry over into this setting as well? If that is too difficult to answer, do you think Procrustes-based similarity metrics also suffer from this systematic bias? It would be nice to see some discussion on this front.

**Ethical Concerns:**

["NO or VERY MINOR ethics concerns only"]

**Final Justification:**

The authors addressed all of my concerns. I am maintaining my score of 5 (accept).

**Limitations:**

Yes.

**Paper Formatting Concerns:**

N/A.

**Quality:**

3

**Strengths And Weaknesses:**

This paper tackles an important question at a good time. It is mostly well-written, and the motivation is very clear.

The main weakness is that it really helps to be a physicist when reading this paper, which I am not (I am coming from dynamical systems / control theory / ML).

---

> ### Author Rebuttal · Authors · 2025-07-31
>
> We thank the reviewer for their positive comments and for raising important concerns. We hope our responses clarify the reviewer’s questions, and we are happy to address any follow-up questions.
>
> **Strength & Weakness:**
>
> We are aware that bracket notation is non-standard in the ML community, but we decided to use it because of how it was formulated in the original paper. We apologize for the inconvenience and will try to include a more detailed introduction to bracket notation in the appendix.
>
> **Questions:**
>
> **1:** This is an excellent question and one of the main limitations of our work. Indeed, it is not clear if this is a good model for neurophysiological recordings. We have several reasons to consider this:
>
> 1. It is analytically easy to study since there is ample work on Wishart matrices.
> 2. It has been debated whether neural data can be regarded as random projections. Often, neural data involves convolution and may potentially mix with each other. Also, large-area recordings such as fMRI automatically act as random projections as they involve averaging many units.
> 3. Neural recordings have been found to be high-dimensional in the sense that individual neurons represent population-level information, hinting that the recorded neurons are already mixed.
>
> Although currently we struggle to generalize this to other sampling methods, we want to mention that this method may potentially be used to check whether the individual units are mixed or not. RMT is a powerful tool that can make precise predictions about how spectral quantities change as a function of sample size. One may employ cross-validation techniques to see if the change in the spectra by adding more neurons matches the RMT prediction and potentially informs practitioners. We leave this point to future work.
>
> **2:** We thank the reviewer for raising this insightful point.
>
> First, we can certainly use our method for estimating the noise ceiling by measuring the similarity between repeated trials of the same neural stimulus. Since both the noise ceiling (self-similarity) and neural similarity monotonically decrease with neuron sampling, it is not immediately clear whether their ratio still remains underestimated.  On the other hand, normalizing with the noise ceiling may also result in inflated similarity scores since it is extremely sensitive to neuron sampling.
>
> While we could not make this analysis due to time constraints, it will certainly be interesting to explore this problem.
>
> Additionally, we aim to extend this theory to random projections in which the matrix elements are correlated instead of independent, as in the Wishart case.
>
> We will discuss these points in detail in a separate **Limitations** section.
>
> **3:** We thank the reviewer for bringing this interesting point to our attention. We certainly intend to extend such RMT analysis to other similarity measures, and will provide a more detailed discussion of these extensions in the future directions section.
>
> While an RMT treatment of DSA itself seems difficult because of its complex nature, we would like to mention that this approach may potentially transfer to comparing delay-embedded representations with CKA when there are limited data points or the timeseries is sparsely sampled. However, we note that this still would not capture the entire non-linear nature of DSA.
>
> Extension to Procrustes-based similarity measures is a very interesting idea!  While CKA depends on the Frobenius norm of the cross-covariance, Procrustes distance relies on its nuclear norm (see Pospisil et al. [Ref a]), and we think it is reasonable to expect that a similar bias also exists in Procrustes distance.
>
> Inspired by the reviewer’s question and the cited papers, we are currently conducting simple experiments on ring attractors with finite sampling and hoping to report empirical results on whether such biases exist in DSA and Procrustes.
>
> [a] Pospisil et.al. Estimating shape distances on neural representations with limited samples

---

> > ### Comment · Reviewer_fmbc · 2025-08-04
> > **Thank you.**
> >
> > I appreciate the authors’ response and their addition of future work plans to the Discussion section, particularly the proposed extensions such as DSA. I will maintain my acceptance score of 5.

---

### Official Review · Reviewer_iYBp · 2025-07-02

**Clarity:** 2
**Significance:** 1
**Originality:** 2
**Rating:** 4
**Confidence:** 5

**Summary:**

This paper analyze the representation similarity in neural networks with limited neurons, which is a common scenario in neuroscience as the recorded neurons are often limited in number. The authors develop a theoretical framework based on Random Matrix Theory to analyze how finite neuron sampling systematically underestimates similarity measures like CCA and CKA. They propose a denoising method to infer population-level similarity from limited samples and validate their approach on both synthetic and real neural data.

**Questions:**

See the `Weaknesses` section.

**Ethical Concerns:**

["NO or VERY MINOR ethics concerns only"]

**Final Justification:**

I recommend borderline accept

**Limitations:**

yes

**Paper Formatting Concerns:**

NIL

**Quality:**

2

**Strengths And Weaknesses:**

## Strengths
1. The research question of this paper is important.
2. The authors develop a theoretical framework based on Random Matrix Theory to analyze the effects of limited neuron sampling on similarity measures, which is a novel approach.

## Weaknesses
**Minor concerns**
1. My primary concern is that applying CCA and CKA to analyze neural representational similarity may be fundamentally inappropriate, as these `representational metrics completely ignore the temporal dynamics of neural activity`. This limitation has been extensively discussed in recent similarity measures literature [1-3], highlighting two critical issues:
- Different dynamical systems can yield similar static representations when sampled at discrete time points, and
- Similar underlying dynamics can manifest as different static representations depending on sampling timing.

    However, both CCA and CKA are incapable of addressing these two scenarios, making the comparison between neural recording potentially misleading.

2. The scalability of the authors' approach can be constrained by its dependence on the spectral decomposition properties specific to CCA/CKA. The method cannot easily extend to:
- Dynamic similarity metrics
- Non-linear similarity metrics
- Information-theoretic approaches
  This raises concerns about the broader impact of this work: if the field is moving away from `representation similarity metrics` toward other directions above, then developing sophisticated corrections for potentially obsolete metrics may have limited long-term value for computational neuroscience.

3. The assumption that neuron sampling can be modelled as `random projections` is mathematically convenient but can be a dangerous oversimplification. In practice, electrode placement is often anatomically constrained, and recorded neurons may be spatially clustered or biased toward certain sub-regions. This could violate the independence assumptions underlying the random matrix theory.

**Minor issues**
1. The notation switches between tildes and hats for population vs. estimated quantities, which could be clarified for consistency.

**Reference**
1. Ostrow, M., Eisen, A., Kozachkov, L., & Fiete, I. (2023). Beyond geometry: Comparing the temporal structure of computation in neural circuits with dynamical similarity analysis. Advances in Neural Information Processing Systems, 36, 33824-33837.
2. Kamiya, S., Kitazono, J., & Oizumi, M. (2024, March). Koopman Operator Based Dynamical Similarity Analysis for Data-driven Quantification of Distance between Dynamics. In ICLR 2024 Workshop on Representational Alignment.
3. Zhang, S., Ye, Z., Yan, Y., Song, Z., Wu, Y., & Wu, J. (2025). KoopSTD: Reliable Similarity Analysis between Dynamical Systems via Approximating Koopman Spectrum with Timescale Decoupling. In Forty-second International Conference on Machine Learning.

---

> ### Author Rebuttal · Authors · 2025-07-31
>
> We thank the reviewer for their thorough review. We hope our response addresses the reviewer’s concerns and look forward to discussing it further.
>
> **Weakness:**
>
> **1&2:** We first note that similarity measures are guided by the symmetries and structure present in the data. For instance, CKA and Procrustes treat two representations as similar if they are related by an orthogonal transformation. Depending on the underlying data-generating process, this may or may not be an appropriate symmetry to assume, and in some cases, alternative metrics may be more suitable.
>
> We acknowledge that both CCA and CKA are static similarity measures and therefore do not account for temporal dynamics or nonlinear structure in the data. However, our work explicitly targets static representations, which remain widely used and scientifically relevant—for example, in studies of early visual cortex, where neurons are often modeled via their time-averaged firing rates. In such settings, the assumption of Poisson-distributed spiking and the use of rate-based representations are standard practice in neuroscience.
>
> Regarding the concern about long-term relevance and impact, while our theoretical analysis focuses on CKA, the core phenomenon we study—sampling-induced spectral bias—is not unique to CKA or even to kernel methods. Any approach that summarizes high-dimensional neural activity using operators such as covariance matrices, Gram matrices, or cross-spectral matrices will exhibit analogous finite-sample biases in their eigenspectra. Thus, we believe our framework lays the foundation for extensions to more complex or dynamic similarity metrics in future work.
>
>
> **3:** We agree that modeling neuron sampling as i.i.d. random projections is a convenient baseline but can be unsuitable when anatomical constraints induce spatial clustering and correlated activity, potentially violating the independence assumptions behind our RMT analysis. Our goal in this paper is to foreground—apparently for the first time—that neuron‑wise sampling itself reshapes the spectrum and can bias CKA/CCA; this effect has not been examined thoroughly. We now state this assumption explicitly and position our results as a baseline. Extending the analysis to structured, non‑i.i.d. sampling (e.g., spatially biased or block‑correlated selections) is an important direction, and we plan to pursue it in follow‑up work.
>
> Please also see our response to Reviewer fmbc Q.1
>
> **Minor issues:**
>
> We thank the reviewer for pointing out this confusion. In our notation, tildes correspond to population quantities; however, we have another quantity,  $\hat{\tilde M}$, which denotes the estimated value of the population $\tilde M$ from sample quantities. We will abandon this notation and instead refer to it as $\tilde M_{est}$.
>
> Please also see our response to Reviewer HcPT Q.4b.

---

> > ### Comment · Reviewer_iYBp · 2025-08-02
> > **Thanks for your response**
> >
> > As I have mentioned in my initial review, different dynamical systems can yield similar static representations when sampled at discrete time points. Moreover, similar underlying dynamics can manifest as different static representations depending on sampling timing.
> >
> > These viewpoints have long been intently overlooked in the computational neuroscience community. Researchers have assumed that representational similarity equates to similarity between dynamical systems (e.g, neuronal level, neural circuits level, brain regions level, or whole-brain level dynamics), since a lack of tools that could directly infer dynamical systems from representations and then measure similarity between these underlying systems [1-4].
> >
> > However, recently proposed dynamics-based similarity metrics [3-4] for assessing dynamical system similarity now allow us to revisit this fundamental issue. In my view, continuing to study the sampling effects of CKA and CCA has become meaningless in the neuroscience context, as these measures are inherently flawed for capturing what we actually care about - the dynamics themselves rather than their static snapshots.
> >
> > ---
> >
> > "i.i.d. random projections" is a dangerous oversimplification. Simply stating they will pursue "structured, non-i.i.d. sampling" in future work sidesteps the immediate concern that their current findings may not generalize meaningfully to the most commonly used neuroimaging modalities like fMRI (BOLD signals have massive spatial smoothing) and EEG (volume conduction effects that violate independence assumptions), where the i.i.d. assumption is fundamentally violated. This will significantly decrease the importance of your method, especially when you apply it to neuroscience field.
> >
> > ---
> >
> > **Reference**
> > 1. Braun, L., Grant, E., & Saxe, A. M. (2025). Not all solutions are created equal: An analytical dissociation of functional and representational similarity in deep linear neural networks. In Forty-second International Conference on Machine Learning.
> > 2. Nejatbakhsh, A., & Wang, Y. Identifying Neural Dynamics Using Interventional State Space Models. (2025). In Forty-second International Conference on Machine Learning.
> > 3. Ostrow, M., Eisen, A., Kozachkov, L., & Fiete, I. (2023). Beyond geometry: Comparing the temporal structure of computation in neural circuits with dynamical similarity analysis. Advances in Neural Information Processing Systems, 36, 33824-33837.
> > 4. Zhang, S., Ye, Z., Yan, Y., Song, Z., Wu, Y., & Wu, J. (2025). KoopSTD: Reliable Similarity Analysis between Dynamical Systems via Approximating Koopman Spectrum with Timescale Decoupling. In Forty-second International Conference on Machine Learning.

---

> > > ### Author Response · Authors · 2025-08-04
> > >
> > > We first want to clarify the intent of this paper. We analyze the limitations of commonly used static similarity measures and propose a principled correction. Our goal is not to claim these metrics are universally sufficient, but to make their use more reliable when they are employed by practitioners.
> > >
> > > RSA/CKA/CCA [a-e] are among the most popular similarity measures in neuroscience. However, as we demonstrated (also see [f]), their naive applications may yield misleading results. Having identified this issue, our work 1) studies when and how things can go wrong and 2) proposes a correction method. Similar to previous works [g], our study urges practitioners to be careful when drawing conclusions from such measures. The fact that linear scores do not capture all aspects of similarity does not mean we should abandon them—rather, we should understand their failure modes and improve them.
> > >
> > > We reiterate that we agree with the reviewer that linear similarity measures can give wrong results if two representations are related by a highly nonlinear transformation. The DSA papers cited by the reviewer illustrate such cases in controlled dynamical systems, where two datasets can be related by an arbitrary diffeomorphism. The main idea of DSA is finding a linear (Koopman) representation of each time series and then measuring their similarity up to a linear transformation. This method identifies two systems if their Koopman representations are equivalent up to a similarity transformation.
> > >
> > > However, we respectfully disagree with the reviewers' assertion that the linear measures are meaningless because we have DSA now. The reviewers base their claim on Refs.[1-4] but these works demonstrate DSA primarily on synthetic or small-scale simulated neural data, and do not definitively conclude their effectiveness on real brain data in comparison to CKA. Advocating for the complete abandonment of CKA/RSA requires strong evidence that neural representations are never linearly related, and should fail 100% of the time on real data. On the contrary, many large-scale studies continue to show that there is an informative structure even with linear metrics [h-j].
> > >
> > > We note that widely used linear measures in neuroscience, such as CKA/RSA, have been shown to yield meaningful results in many studies. While they may fail to describe nonlinear aspects of the data, this does not make them useless, as the neural data also show linear relationships. This point of view completely undermines countless neuroscience research studies that sought linear relationships between different neural systems and extracted meaningful patterns in data [k]. Linear measures remain useful even if they do not capture every non-linear aspect, much like PCA remains an essential tool for data analysis despite its linearity.
> > >
> > > Finally, we want to list several arguments that justify the relevance of those methods despite their inability to capture non-linear properties.
> > >
> > > - Our methods are not limited to neuroscience alone. As our title suggests, they aim at representational similarity as a whole. As shown in Fig.1, it applies to artificial neural networks, too.
> > >
> > > - Beyond being a model for how neural recordings are sampled, random projections also serve as a data analysis method. They enable effective subsampling from arbitrary populations without distorting the geometry, making them extremely useful for analyzing large datasets. In this case, our analysis shows that the subsampled population may strongly deviate from the original population depending on the level of sampling.
> > >
> > > - The reviewers’ critique primarily applies to neural systems with strong dynamical properties. However, in certain contexts, static similarity metrics remain informative, such as neuron recordings from early visual cortex in response to static images [h-j].
> > >
> > > - As we mentioned before, a major neural data analysis technique is modeling individual neurons as parametric random processes (e.g., Poisson). Once each neuron model is fit to data, the resulting parameters (e.g., firing rates) may indeed act as static representations. Unlike the reviewers suggest, these are not snapshots since these parameters describe the entire dynamics.
> > >
> > > - In certain cases, representations may dynamically change in a way that preserves the sample-wise similarity. This is the case for representational drift [l,m].
> > >
> > > In summary, this work is part of a larger research program that aims to develop analytical theories of common similarity measures (including DSA) to understand their robustness. We appreciate the reviewer’s push towards dynamical similarity metrics; however, we believe the present work is a necessary stepping stone: understanding and correcting sampling bias in the static limit provides the foundation on which a thorough dynamical theory can be built.

---

> ### Author Response · Authors · 2025-08-04
>
> [a] Kriegeskorte et al. (2008) “Representational similarity analysis—connecting the branches of systems neuroscience.” Frontiers in Systems Neuroscience.
>
> [b] Nili et al. (2014) “A Toolbox for Representational Similarity Analysis.” PLoS Comp. Bio.
>
> [c] Kornblith et.al. (2019) “Similarity of Neural Network Representations Revisited (CKA).” ICML.
>
> [d] Morcos et.al. (2018) “Insights on representational similarity in neural networks with CCA.”
>
> [e] Sucholutsky et al. (2023) “Getting aligned on representational alignment.”
>
> [f] Murphy et.al. (2024) “Correcting Biased Centered Kernel Alignment Measures in Biological and Artificial Neural Networks”
>
> [g] Canatar et.al. (2023) “A spectral theory 346 of neural prediction and alignment.” NeurIPS.
>
> [h] Yamins et.al. (2014) “Performance-optimized hierarchical models predict neural responses in higher visual cortex.” PNAS.
>
> [i] Khaligh-Razavi and Kriegeskorte. (2014) “Deep Supervised, but Not Unsupervised, Models May Explain IT Cortical Representation”. PLoS Comp. Bio.
>
> [j] Schrimpf et.al. (2018) “Brain-score: Which artificial neural network for object recognition is most brain-like?”
>
> [k] Kar and DiCarlo. (2024) “The Quest for an Integrated Set of Neural Mechanisms Underlying Object Recognition in Primates.” Ann. Rev. Vis. Sci.
>
> [l] Deitch et al. (2021) “Representational drift in the mouse visual cortex.” Curr. Biol.
>
> [m] Roth and Merriam. (2023) “Representations in human primary visual cortex drift over time.” Nat. Comm.

---

> > ### Comment · Reviewer_iYBp · 2025-08-08
> > **Thanks for your additional response and clarification**
> >
> > Unlike reviewer HcPT and the authors, I maintain my views on the dangers of the independent and identically distributed assumptions for the proposed method in neuroscience applications since there is no idealized world (in response to reviewer HcPT's claim), right?
> >
> > I also doubt the effectiveness of CKA/CCA in measuring the similarity of different neural recording data. I raise this question because this manuscript is submitted to the neuroscience track, yet some papers have recently noticed the reliability of representation-based similarity measurements in this field.
> >
> > However, I believe the authors' method is indeed helpful in understanding and correcting sampling bias in the static limit. I think the researchers from the other field may benefit from the opinions in this work.
> >
> > I decide to increase my score to 4.

---

> > > ### Comment · Reviewer_HcPT · 2025-08-08
> > >
> > > To be very clear, I wholeheartedly agree with you that the idealized Gaussian world does not exist. Where we diverge is in the question of whether results derived imagining that we lived in a Gaussian paradise can give useful insights into our non-Gaussian reality. The case study I have in mind here is kernel ridge regression, where it turns out that Gaussian universality of the test risk holds fairly broadly---see for instance [Misiakiewicz and Saeed (2024)](https://arxiv.org/abs/2403.08938) or [Wei et al. (2022)](https://proceedings.mlr.press/v162/wei22a.html)---and from the other end there is a line of work trying to more systematically determine under what conditions of the data these predictions fail, see e.g. [Tomasini et al. (2022)](https://proceedings.mlr.press/v162/tomasini22a.html). The Gaussian computation is not the end of the story, but is a useful starting point.

---

> > > > ### Comment · Reviewer_iYBp · 2025-08-09
> > > >
> > > > Thanks for the clarification. I believe I now understand your point. I’ll go through the reference you shared carefully. Cheers.

---

### Official Review · Reviewer_BZoy · 2025-07-03

**Clarity:** 3
**Significance:** 3
**Originality:** 3
**Rating:** 5
**Confidence:** 3

**Summary:**

Measuring representational similarity between computational models and neural recordings has become a staple methodology in computational neuroscience. In practice, although we can access the entire population of neurons in computational models, we can record only a limited number of neurons in the brain, hoping they satisfactorily represent the larger population. In this work, the authors apply Random Matrix Theory to investigate how this sampling limitation impacts similarity measures. They find that similarity scores from popular measures are systematically underestimated under finite neuron sampling, and they propose a method to infer the population similarity. Theoretical predictions are validated on both synthetic and real datasets.

**Questions:**

- Fig. 2: How many sampling is done per neuron size N?
- Why is figure 5 used before figure 4 in the text?
- The proof that CKA or CCA are underestimated differently across models is strong enough evidence to motivate the usefulness of the proposed correction. That being said, it would have further strengthened the paper to push further the empirical investigation in section 5.2 to see if any of the previously established insights change when applying the proposed correction.

**Ethical Concerns:**

["NO or VERY MINOR ethics concerns only"]

**Final Justification:**

I have read all the reviews and discussions. I did not have major concerns, and while some reviewers have raised important limitations (scope), the authors have convincingly addressed them. I believe this paper deserves to be accepted.

**Limitations:**

Yes.

**Paper Formatting Concerns:**

/

**Quality:**

3

**Strengths And Weaknesses:**

### Strengths
- This paper tackles an important problem and contributes a practically useful methodology for the community. Representational similarity metrics are broadly used to infer good computational models of the brain, and we need to be able to trust those measures despite the practical limitations when recording neural activities.
- The paper is well written and well illustrated.
- The investigation carried out in the paper is thorough and convincing.

### Weaknesses
- I do not see any major weakness in this paper.

---

> ### Author Rebuttal · Authors · 2025-07-31
>
> We thank the reviewer for their kind review and hope our responses addressed the reviewer’s questions.
>
> **Questions:**
>
> **1:**  We thank the reviewer for catching this. In Fig.2, we had 20 trials of sampling neurons, and we will mention it in the caption.
>
> **2.** We are sorry for the confusion. We have fixed the organization now.
>
> **3.** We thank the reviewer for this suggestion and agree that application to other data would be valuable. We are actually aiming to release a codebase that can be generalized to any dataset. We will also try to include some new experiments on additional datasets in our revised manuscript.

---

> > ### Comment · Reviewer_BZoy · 2025-08-07
> >
> > Thank you for answering my questions.
> >
> > I have read the reviews and the interesting discussions that took place between the authors and the different reviewers.  Similar to Reviewer HcPT, I think that while some of the limitations raised are fair, I do not agree that they invalidate the work done in this paper. It has simply highlighted that the paper would be strengthened by a limitation section.
> > And in general, I think the push back by the other reviewers forced the authors to better contextualize the relevance of their work (e.g., while dynamics-based similarity metrics might be more accurate in the long term, as of now CKA are still broadly used and it clearly captures meaningful signal, or the fact that yes the title is broad but the paper is indeed heavily geared toward neuroscience yet the framework/insight are just as interesting to the ML community) and the paper would be clearer with those contextualization.
> >
> > Overall, I will keep my current recommendation to accept this paper.

---

### Decision · Program_Chairs · 2025-09-17

**Decision:**

Accept (poster)

**Comment:**

This submission studies the behavior of CCA and CKA when subsampling neurons under the assumption that the subsampling happens through a Gaussian random projection of the population activity and the eigenvalues of the population Gram matrix exhibit power-law decay. It proposes an algorithm to infer the population similarity values from limited data, evaluates it in a synthetic setting, demonstrating that it can draw accurate conclusions where the naive estimator would fail, and analyzes its behavior on brain data.

Reviewers fmbc and BZoy were initially supportive of acceptance and had only minor concerns. Reviewer HcPT was initially concerned that the proposed method is heuristic and its bias is difficult to characterize. While the authors agreed that the bias is difficult to characterize, they showed that it is asymptotically unbiased and gave a method to derive confidence intervals, which the reviewer found convincing. Reviewer iYBp felt that the enterprise of measuring representational similarity was itself flawed since static representational similarity measures have limitations when applied to dynamical systems such as the brain, and the assumption of subsampling by random projections may not be accurate if the units of subsampling are actually axis-aligned neurons. Although these are both valid concerns, as Reviewer HcPT notes, these representational similarity measures are widely used and there exist problems where they do give useful insight, and the assumption of Gaussianity could still give useful results in a non-Gaussian world. Reviewer iYBp was ultimately convinced and raised their score to borderline accept.

Overall, this submission provides a novel and theoretically-justified (if still somewhat heuristic) algorithm to address bias when measuring similarity between subsampled representations. The underlying theory is interesting, and the algorithm is clearly better than naive estimation, making it likely that experimentalists will find it valuable for practical applications. I thus support its acceptance.